# Proximal CA1 20–40 Hz power dynamics reflect trial-specific information processing supporting nonspatial sequence memory

Sandra Gattas[1,2], Gabriel A Elias[2,3], John Janecek[2,3], Michael A Yassa[2,3]*, Norbert J Fortin[2,3]*

[1]Department of Electrical Engineering and Computer Science, University of California, Irvine, United States; [2]Center for the Neurobiology of Learning and Memory, University of California, Irvine, United States; [3]Department of Neurobiology and Behavior, University of California, Irvine, United States

**Abstract** The hippocampus is known to play a critical role in processing information about temporal context. However, it remains unclear how hippocampal oscillations are involved, and how their functional organization is influenced by connectivity gradients. We examined local field potential activity in CA1 as rats performed a nonspatial odor sequence memory task. We found that odor sequence processing epochs were characterized by distinct spectral profiles and proximodistal CA1 gradients of theta and 20–40 Hz power than track running epochs. We also discovered that 20–40 Hz power was predictive of sequence memory performance, particularly in proximal CA1 and during the plateau of high power observed in trials in which animals had to maintain their decision until instructed to respond. Altogether, these results provide evidence that dynamics of 20–40 Hz power along the CA1 axis are linked to trial-specific processing of nonspatial information critical to order judgments and are consistent with a role for 20–40 Hz power in gating information processing.

*For correspondence:
michael.yassa@uci.edu (MAY);
norbert.fortin@uci.edu (NJF)

**Competing interest:** The authors declare that no competing interests exist.

## Editor's evaluation

This article presents intriguing evidence that 20–40 Hz amplitude increases in the hippocampus are tied to task-relevant parameters, namely, odors presented in a sequence, as well as learning. The results reveal new insights about hippocampal processing of nonspatial information and contribute to a greater understanding of hippocampal network mechanisms of memory processing.

## Introduction

Brain oscillations are associated with many cognitive functions (*Buzsáki et al., 2013*; *Colgin, 2016*; *Pesaran et al., 2018*) and are thought to reflect complex interactions of neural activity from diverse populations of interconnected neurons (*Pesaran et al., 2018*). For instance, it is well established that the hippocampus is critical for spatial learning and memory (*O'Keefe and Nadel, 1978*) and that distinct oscillatory states are observed in hippocampal subregion CA1 during spatial navigation in rodents (see *Colgin, 2016*). CA1 theta power is prominent during spatial exploration, increases with running speed, and its phase is thought to link representations of spatial locations within a trajectory (*Buzsáki and Moser, 2013*; *Drieu and Zugaro, 2019*). CA1 also exhibits transient increases in slow (25–55 Hz) and fast gamma (60–100 Hz) power during exploration, which are thought to be associated with retrieval and encoding processes, respectively (*Colgin et al., 2009*). It remains unclear, however,

the degree to which the aforementioned frequency content and associated functions observed during spatial exploration generalize to nonspatial forms of information processing.

In addition to spatial information, accumulating evidence indicates that the hippocampus plays a key role in the processing of temporal information. Consistent with its unique architecture and connectivity (*McNaughton and Morris, 1987*; *Lisman, 1999*; *Foster and Knierim, 2012*; *Buzsáki and Tingley, 2018*), a growing literature shows that the hippocampus is critical for remembering sequences of nonspatial events (*Fortin et al., 2002*; *Kesner et al., 2002*; *Allen and Fortin, 2013*; *Eichenbaum, 2014*) and hippocampal neurons code for temporal relationships among such events (*MacDonald et al., 2011*; *Allen et al., 2016*; *Shahbaba et al., 2022*). However, little is known about the oscillatory dynamics associated with this fundamental type of information processing in the hippocampus. Previous studies have observed theta oscillations in CA1 while rodents sampled task-relevant nonspatial stimuli, though the frequency range tended to be lower than during running. For example, ~4–8 Hz oscillations are observed in tasks using olfactory stimuli (*Martin et al., 2007*; *Igarashi et al., 2014*; *Allen et al., 2016*). In addition, the hippocampus exhibits oscillations in the 20–40 Hz range during the processing of odor information (*Martin et al., 2007*; *Kay, 2014*; *Igarashi et al., 2014*; *Allen et al., 2016*), with this signal varying in power across the proximodistal axis of CA1 (higher power in distal than proximal CA1; *Igarashi et al., 2014*). Notably, the 20–40 Hz frequency range overlaps with slow gamma (~25–55 Hz) as well as with beta (~15–35 Hz) and its functional relevance remains poorly understood. Based on findings from striatal and cortical circuits (*Leventhal et al., 2012*; *Engel and Fries, 2010*; *Schmidt et al., 2019*), beta has been hypothesized to serve a 'gating' function and reflects a 'status quo' signal bridging the period between a decision and a response. It is unclear whether this function extends to the role of 20–40 Hz in hippocampus and, more generally, how the recruited spectral profiles in CA1 support memory for temporal order.

To address these issues, we examined CA1 local field potential (LFP) recordings while rats performed a hippocampus-dependent, odor-based sequence memory task (*Fortin et al., 2016*). This task offers precise time-locking to stimulus presentations and responses, as well as different trial types associated with distinct cognitive demands. As in our previous work (*Allen et al., 2016*), we report power increases in the 20–40 Hz and low-theta frequency ranges during the odor sequence processing periods. Here, we extend these results by demonstrating that the same electrodes exhibited distinct spectral content in a different state (running on the track), which was characterized by high-theta power and a broadband increase in power for frequencies above 24 Hz. Notably, power in the recruited frequencies varied along the proximodistal axis of CA1 in a behavioral state-dependent manner. Power in the 20–40 Hz range increased with session performance, whereas power in other frequency ranges did not. This performance effect was primarily driven by activity on trials involving a *sustained hold response*. More specifically, we found that 20–40 Hz power gradually increased during odor presentations and reached a plateau (steady state) on these 'hold-response' trials, and that this steady-state power in the proximal segment of CA1 was key to predicting trial accuracy. Collectively, these results suggest that 20–40 Hz power dynamics in the proximal segment of CA1 are linked to processing the temporal context of nonspatial events and reflect a post-decision state that may serve to protect local, task-critical information processing from disruption and interference until a response can be performed.

## Results

Five rats were trained on an odor-based sequence memory task that critically depends on the hippocampus (*Fortin et al., 2016*). In this task, animals receive repeated presentations of a sequence of five odor items and must correctly identify each item as being presented 'in sequence' (InSeq; e.g., AB<u>C</u>…) or 'out of sequence' (OutSeq; e.g., AB<u>E</u>…) to receive a water reward (*Figure 1A and B*). More specifically, each odor presentation is triggered by the rat's nose entering the odor port and the duration of the nosepoke response is used to behaviorally indicate whether the presented item is thought to be InSeq (by continuing to hold until an auditory signal occurs at 1.2 s) or OutSeq (by withdrawing before the signal). Note that the exact time cutoff used to classify InSeq and OutSeq responses was determined separately per animal based on the intersection of their distributions of nosepoke durations on InSeq and OutSeq trials (*Figure 1G*; ~1.1 s). This is to account for the observation that animals occasionally withdrew just before the 1.2 s threshold (because of the predictable timing of the auditory stimulus), suggesting that these are anticipatory errors not incorrect InSeq/OutSeq

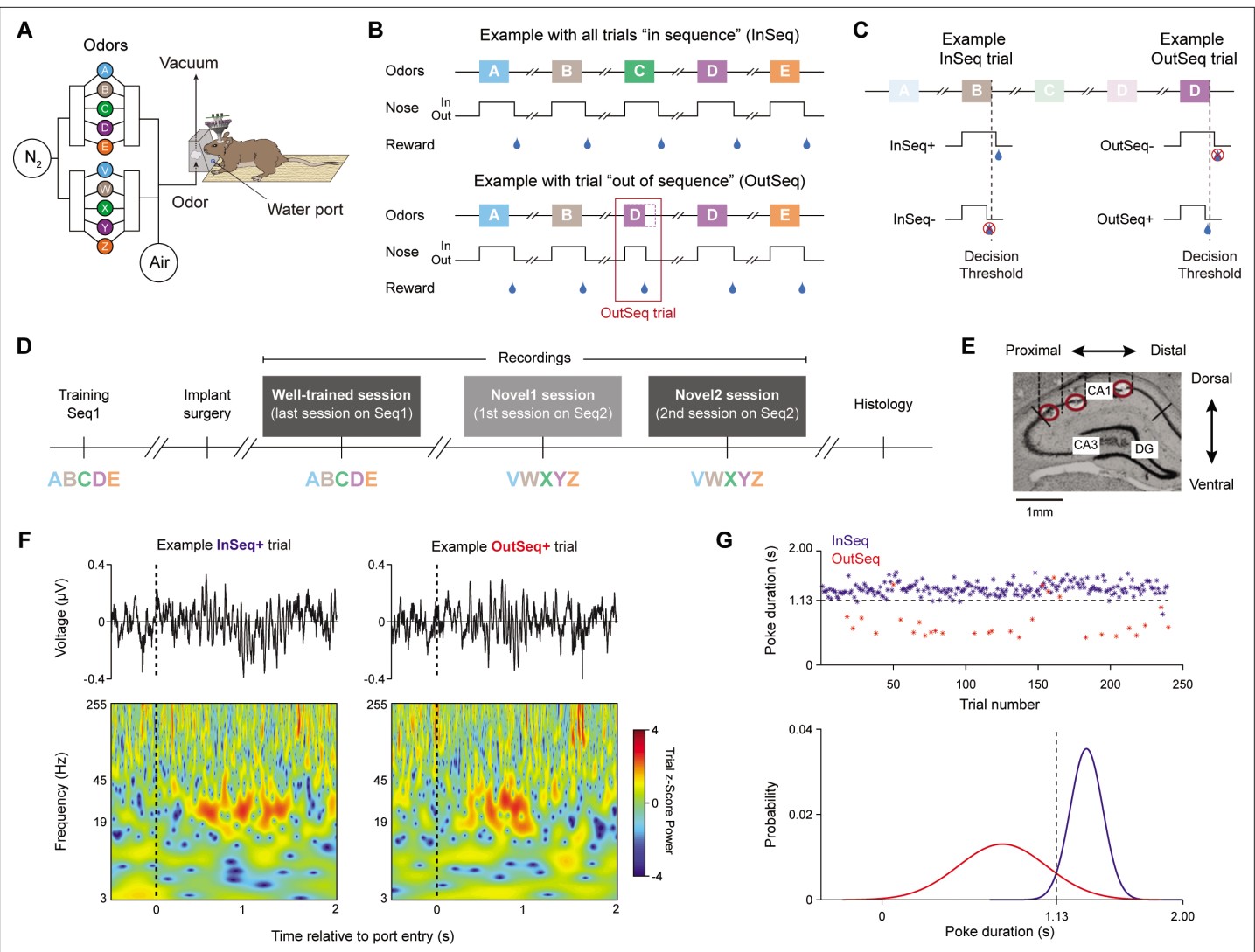

**Figure 1.** Local field potential (LFP) activity was recorded from hippocampal region CA1 as rats performed a nonspatial sequence memory task. (**A**) Using an automated odor delivery system located at one end of a linear track, rats were presented with a series of five odors delivered in the same odor port. (**B**) In each session, the same sequence of five odors was presented multiple times. Approximately half the presentations included all items 'in sequence' (InSeq; e.g., ABCDE; top) and the other half including one 'out of sequence' item (OutSeq; e.g., ABDDE; bottom). For each odor presentation, rats were required to behaviorally indicate whether the presented odor was InSeq (by holding their nosepoke response until the auditory signal at 1.2 s) or OutSeq (by withdrawing their nose before the signal; <1.2 s) to receive a water reward. After completion of each sequence (correctly or incorrectly), animals were required to run to the opposite end of the linear track and return to the odor port before the next sequence could be presented. (**C**) For InSeq items, hold durations *longer* than the decision threshold were considered correct responses (InSeq+) by the hardware and shorter ones incorrect responses (InSeq-). For OutSeq items, hold durations *shorter* than the decision threshold were considered correct responses (OutSeq+) and longer ones incorrect responses (OutSeq-). (**D**) Data analysis focused on LFP activity recorded from three sessions: a session in which animals were tested on the sequence learned preoperatively in which they performed at a high level (ABCDE; well-trained session), as well as two successive sessions testing a novel sequence (VWXYZ; novel 1 and novel 2 sessions) in which they performed at low and moderate levels, respectively. (**E**). Sample coronal section (~4 mm posterior of bregma) showing the range of tetrode tip locations, which spanned much of the proximodistal axis of CA1 (three tip locations shown; red circles). For each animal, a set of four tetrodes equally distributed across the proximodistal axis (with comparable locations across animals) was used for local field potential activity analysis. (**F**) Representative LFP traces (top) and corresponding time-frequency plots (bottom) from example InSeq+ and OutSeq+ trials. (**G**) Nosepoke durations on InSeq and OutSeq trials from one representative animal (well-trained session; top). Normal distribution functions were fit separately for observed InSeq and OutSeq trials (bottom) and the intersection of the two distributions (1.13 s in this case) was used as the time cutoff to classify 'hold' and 'withdraw' responses during analysis (to account for anticipation errors occurring a few ms before the decision threshold). Panels from this figure were adapted from our previous work (*Allen et al., 2016*; *Shahbaba et al., 2022*).

decisions. Importantly, this slight adjustment in cutoffs (~1.1 s instead of 1.2 s) more accurately reflects the animals' decisions but does not change the pattern of results observed. For InSeq items, hold durations *longer* than the identified time cutoff were categorized as correct responses (InSeq+) and shorter ones as incorrect responses (InSeq-; *Figure 1C*). The converse was true for OutSeq items: hold durations *shorter* than the time cutoff were categorized as correct responses (OutSeq+) and longer ones as incorrect responses (OutSeq-).

Following a training period of 2–4 months, animals were surgically implanted with a microdrive over the dorsal CA1 region of the hippocamps and tetrodes were then slowly driven to the pyramidal layer over a period of 2–3 weeks. LFP activity was then recorded from tetrodes spanning much of the CA1 proximodistal axis as animals performed the task (*Figure 1D–F*). We analyzed LFP activity from three sessions: a session in which animals were tested on the sequence learned preoperatively (ABCDE; well-trained session; n = 5 rats), as well as two successive sessions testing a novel sequence (VWXYZ; novel 1 and novel 2 sessions; n = 4 rats, as one animal damaged its microdrive after the well-trained session). According to our previously established performance metric (sequence memory index [SMI]; *Allen et al., 2014*), behavioral accuracy in these sessions ranged from low (novel 1; SMI = 0.0619 ± 0.0664; mean ± SEM), to moderate (novel 2; SMI = 0.2624 ± 0.0671), to high (well-trained; SMI = 0.5906 ± 0.0578), providing an opportunity to compare LFP activity across three different levels of performance. Note that the SMI ranges from 0 to 1 and takes into account the uneven distribution of trial types (~90% of trials are InSeq), which makes it a better indicator of performance than percent correct. For additional perspective, the SMI of 0.59 observed in the well-trained session represents asymptotic task performance that corresponds to 92.54 and 68.09% correct on InSeq and OutSeq trials, respectively. In contrast, performance in the novel 1 session is not different from chance (SMI of 0), which is defined by a response pattern following the probabilities of each trial type (in this case, ~90% of trials were InSeq and subjects were 88.25 and 17.47% correct on InSeq and OutSeq trials, respectively). Performance in the novel 2 session is significantly above that of novel 1 but below that of well-trained (92.16 and 31.87% correct on InSeq and OutSeq trials).

## Odor sequence processing and running epochs are associated with distinct spectral features, which vary across the CA1 proximodistal axis

We began by investigating whether distinct spectral features in CA1 accompany odor sequence and running periods of the task, and whether the observed spectral content varied across the proximodistal axis of CA1. To do so, group (n = 5) peri-event spectrograms were generated from four electrode locations along the proximodistal axis and aligned to odor processing (*Figure 2A*) and running (*Figure 2B*) epochs. For each electrode site, spectrograms were generated using analytic Morlet wavelets and spectral power was z-score normalized to power at the same site during a 30 min period of the recording session. Data were taken from the well-trained session in which animals performed at a high level. For all group analyses, a random intercept linear mixed-effects model (LMEM ANOVA) was fit to the data where electrodes, sessions, or trial types were treated as fixed effects and subjects were treated as a random effect (to account for inherent correlations from repeated measures at the subject level; see Materials and methods for model formulation). For all analyses, data from the same trial type (e.g., InSeq+) was collapsed across sequence positions because no clear modulation of power by position was observed (e.g., *Figure 2—figure supplement 1*).

We found that power in the 20–40 Hz and theta (4–12 Hz) range observed during odor presentations showed distinct patterns across the proximodistal axis (*Figure 2A, C and D*; both frequency ranges defined a priori). More specifically, 20–40 Hz power (110–1200 ms period following port entry) increased toward proximal CA1 (LMEM ANOVA significant effect of electrode: $F_{3,2214} = 41.995$, p<0.001). Power in the most proximal site (electrode 1) showed significantly higher power compared to the two most distal sites (electrodes 3 and 4) ($t_{2214} = -8.072$, p<0.001 and $t_{2214} = -8.024$, p<0.001, respectively). Power in electrode 1 did not differ significantly from that of electrode 2 ($t_{2214} = -0.206$, p=0.837). In contrast, theta power in the same period increased toward distal CA1 (LMEM ANOVA significant effect of electrode: $F_{3,2214} = 60.952$, p<0.001). Electrodes 2–4 showed significantly higher theta power compared to electrode 1 ($t_{2214} = 5.008$, p<0.001, $t_{2214} = 11.99$, p<0.001, and $t_{2214} = 10.733$, p<0.001, respectively).

The same electrodes exhibited a different spectral pattern during running periods, which occurred between sequence presentations (*Figure 2B, E and F*). More specifically, the 20–40 Hz power observed

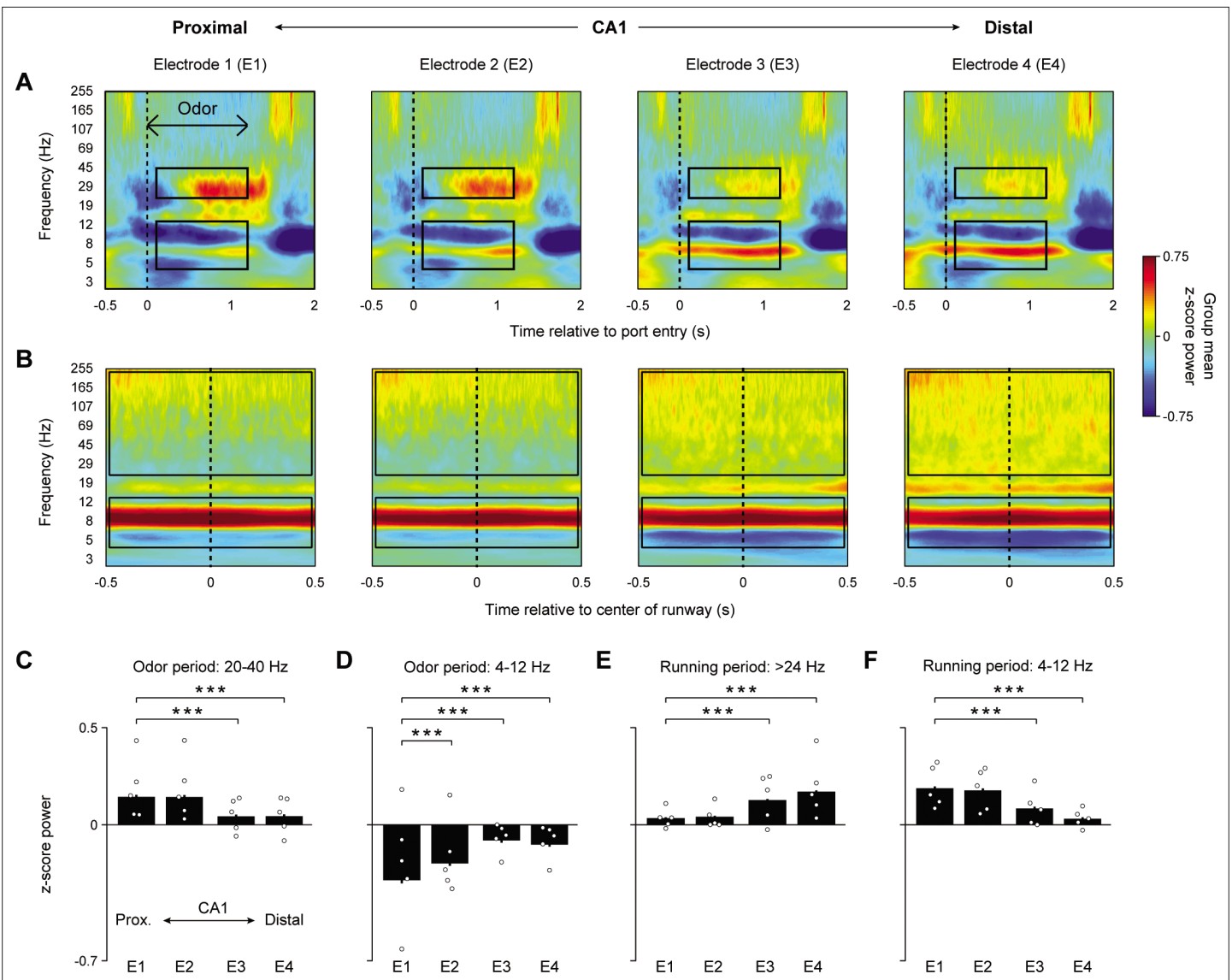

**Figure 2.** Odor sequence processing and running are associated with distinct CA1 oscillatory states, which vary across the proximodistal axis. (**A**) Group peri-event spectrograms (n = 5) during odor sampling period in four electrode locations along the CA1 proximodistal axis (InSeq+ trials only; 0 ms = port entry). Black boxes indicate the time-frequency periods (defined a priori) examined in (**C**) and (**D**). (**B**) Group peri-event spectrograms from the same electrodes during the running period (0 ms = center of the runway). Black boxes refer to the time-frequency periods examined in (**E**) and (**F**). (**C**) Mean z-score 20–40 Hz power during 110–1200 ms period of odor presentation significantly increased toward proximal CA1 (significant effect of electrode: $F_{3,2214}$ = 41.995, p<0.001). (**D**) Same as in (**C**), but for theta (4–12 Hz) showing increased power in the opposite direction (toward distal CA1; significant effect of electrode: $F_{3,2214}$ = 60.952, p<0.001). (**E**) Mean z-score higher frequency power during running (>24 Hz to avoid theta's first harmonic) increases toward distal CA1 (significant effect of electrode: $F_{3,2768}$ = 181.34, p<0.001). (**F**) Same as in (**E**), but for theta showing increased power in the opposite direction (toward proximal CA1; significant effect of electrode: $F_{3,2768}$ = 71.005, p<0.001). Bar plots display mean ± SEM, with individual subject means overlaid as circles. All data were extracted from the well-trained recording session (high-performance level). ***p<0.001 two-tailed t-tests (Bonferroni-corrected).

The online version of this article includes the following figure supplement(s) for figure 2:

**Figure supplement 1.** CA1 20–40 Hz power as a function of position in the sequence.

during odor sampling was weak during running. Additionally, contrary to the odor processing periods, theta power during running was significantly higher in proximal CA1 (LMEM ANOVA significant effect of electrode: $F_{3,2768}$ = 71.005, p<0.001; *Figure 2F*) and was characterized by a higher center frequency (~9 Hz during running compared to ~6 Hz during odor sampling). Electrode 1 showed significantly higher power compared to electrodes 3 and 4 ($t_{2768}$ = –8.173, p<0.001, and $t_{2768}$ = –12.414, p<0.001, respectively) while power in electrode 1 did not differ significantly from that of electrode 2 ($t_{2768}$ =

–0.876, p=0.381). The running period was also characterized by increased high-frequency broad-band power (>24 Hz to avoid theta's first harmonic), which significantly increased toward distal CA1 (LMEM ANOVA significant effect of electrode: $F_{3,2768}$ = 181.34, p<0.001; *Figure 2E*). Electrodes 3 ($t_{2768}$ = 13.255, p<0.001) and 4 ($t_{2768}$ = 19.53, p<0.001) showed significantly higher power compared to electrode 1. Power in electrode 2 did not differ significantly from that of electrode 1 ($t_{2768}$ = 1.051, p=0.293).

## CA1 20–40 Hz power increases with session performance

We then examined whether power in the 20–40 Hz range was linked to performance and whether this association varied along the proximodistal axis. To do so, we extended the previous analyses, which were applied to the well-trained session, to two consecutive sessions in which animals learned a novel odor sequence (correctly identified InSeq trials only). This allowed us to compare power across three sessions characterized by low (novel 1, first session on novel sequence), moderate (novel 2, second session on novel sequence), and high levels of performance (well-trained session; *Figure 3*). Unlike the previous analyses, here we evaluated 20–40 Hz power during the 250 ms time window preceding the withdrawal response to capture the period of high power observed toward the end of odor presentations (at which point power has plateaued; *Figure 3B*).

We found that 20–40 Hz power significantly increased with performance level (LMEM ANOVA significant effect of session: $F_{2,4812}$ = 46.71, p<0.001), which complements our previous study showing learning-related differences in waveform amplitude (*Allen et al., 2016*). When collapsing across electrodes, the well-trained session showed significantly higher power compared to novel 1 ($t_{4812}$=8.451, p<0.001), while power during the novel 1 and novel 2 sessions did not significantly differ ($t_{4812}$=1.470, p=0.14168). Notably, this performance effect did not significantly vary across the proximodistal axis (nonsignificant interaction between electrode and session $F_{6,4806}$ = 0.745, p=0.614), but instead scaled with the local amplitude of 20–40 Hz, suggesting that this behavioral correlate is present throughout dorsal CA1. With regard to the main effect of electrode (collapsing across sessions), we observed a significant difference across sites (LMEM ANOVA: $F_{3,4812}$ = 52.784, p<0.001). More precisely, average power was significantly higher in electrode 1 compared to electrodes 2–4 ($t_{4812}$ = –2.969, p=0.003, $t_{4812}$ = –8.896, p<0.001, and $t_{4812}$ = –11.095, p<0.001, respectively) (*Figure 3C*). It is important to note that we chose an a priori time window (−250–0 ms prior to port withdrawal) to quantify this performance effect. We also used a data-driven approach to identify epochs during which 20–40 Hz dynamics statistically vary with performance level (*Figure 3B*), which showed that the performance effect was not limited to the window defined a priori.

Lastly, we tested whether the observed performance-related effects are specific to 20–40 Hz dynamics or generalize to other bands, including theta (4–12 Hz), the slow gamma range nonoverlapping with 20–40 Hz (40–55 Hz), and fast gamma (65–100 Hz; *Figure 3—figure supplement 1*). We found that performance-related effects were weaker, and the patterns differed from those of the 20–40 Hz range. Whereas 20–40 Hz power increased with performance, theta and fast gamma power decreased (theta LMEM ANOVA: $F_{2,4806}$ = 7.681, p<0.001; fast gamma LMEM ANOVA: $F_{2,4806}$ = 8.147, p<0.001) and slow gamma showed a nonlinear relationship with session performance (LMEM ANOVA: $F_{2,4815}$ = 5.993, p=0.003). This suggests that the 20–40 Hz band is unique in showing a distinctive increase in power with performance level.

## 20–40 Hz steady-state power in proximal CA1 is predictive of accuracy for temporal order judgments

To shed light on the type of processing reflected by 20–40 Hz oscillations, we took advantage of the four different trial types included in our paradigm: InSeq trials that were correctly or incorrectly identified (InSeq+, InSeq-) and OutSeq trials correctly or incorrectly identified (OutSeq+, OutSeq-) (*Figure 1C*). We began by visualizing power dynamics as a function of the entire time course of trials, focusing on the electrode and session with highest 20–40 Hz power (electrode 1, well-trained session). Aligning data to port entry (*Figure 4A*, left), we observed that the rising phase (~0–750 ms) was similar across trial types and that differences emerged later in the trial. Specifically, we observed that power was sustained (reaching steady-state) on trials in which animals performed a 'hold' response (InSeq+ and OutSeq-), whereas it fell off earlier (during the transient phase) on trials in which the animals performed a 'withdraw' response (OutSeq+ and InSeq-). Aligning data to port withdrawal confirmed

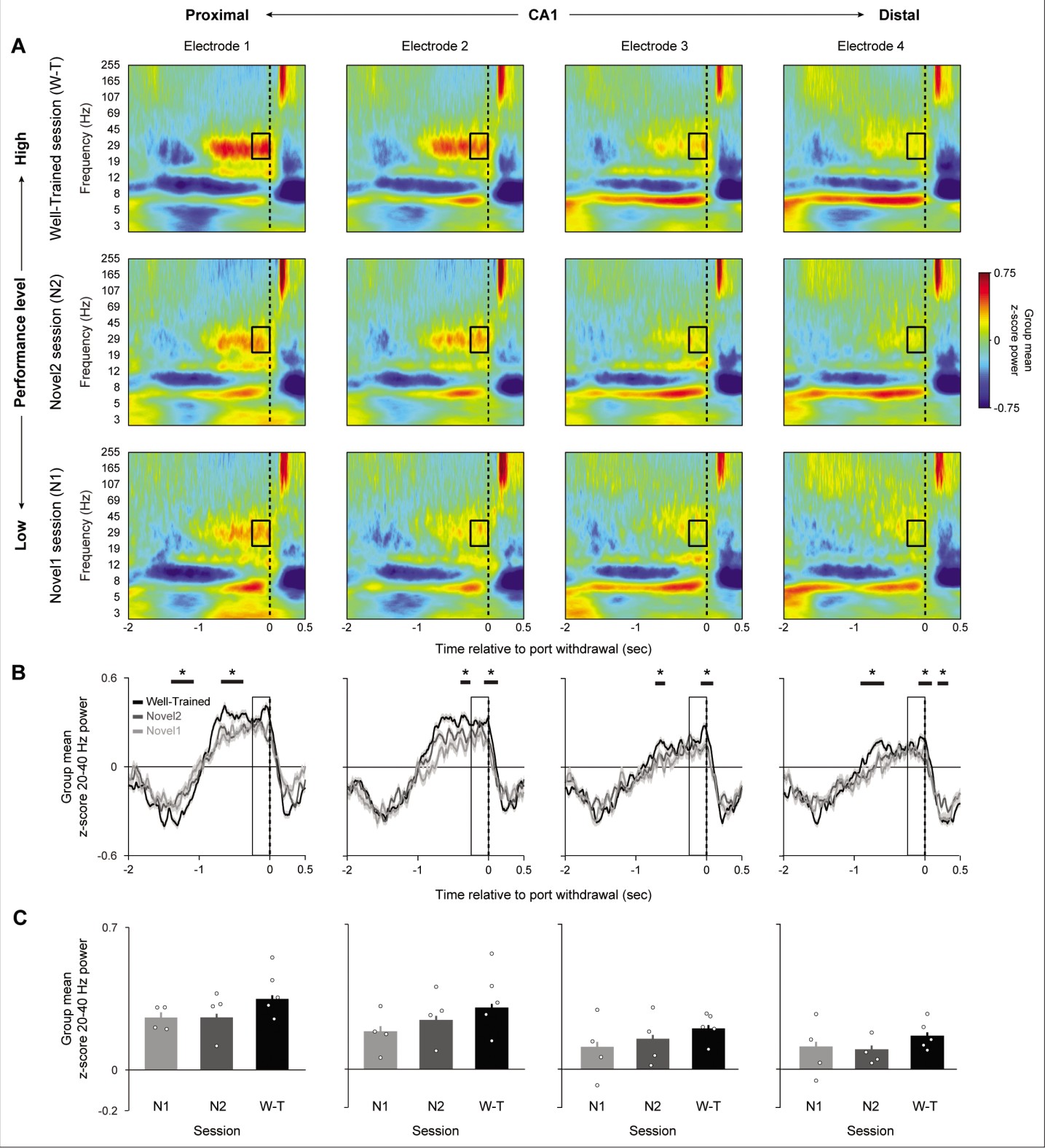

**Figure 3.** CA1 20–40 Hz power increases with knowledge of the sequence. Group peri-event spectrograms across three sessions in which performance levels were low (novel 1, first session on novel sequence; n = 4 rats; bottom row), moderate (novel 2, second session on novel sequence; n = 4; middle row), and high (well-trained session; n = 5; top row). All plots are aligned to port withdrawal (0 ms = port withdrawal) and only include InSeq+ trials. Black box indicates the 250 ms window, defined a priori to capture the 20–40 Hz power plateau, which is analyzed in (**C**). (**B**) Mean z-score 20–40 Hz power traces (± SEM) from data in (**A**). Horizontal black bars indicate epochs with significant performance effect identified using a data-driven approach

*Figure 3 continued on next page*

*Figure 3 continued*

(cluster-based permutation testing [CBPT], n = 1000 permutations). (**C**) Mean z-score power in the 250 ms period prior to port withdrawal (indicated by boxes in **A** and **B**) in each of the three sessions, shown separately for each CA1 site. Power increased with session performance and toward the most proximal sites (significant effect of session: $F_{2,4806}$ = 17.387, p<0.001; significant effect electrode: $F_{3,4806}$ = 8.9081, p<0.001). Bar plots display mean ± SEM, with individual subject means overlaid as circles. *Significant epochs identified in CBPT analysis (p<0.05, corrected for multiple comparisons).

The online version of this article includes the following figure supplement(s) for figure 3:

**Figure supplement 1.** Frequency bands outside the 20–40 Hz range do not show increased power with session performance.

that the withdrawal response occurs during steady-state for the hold responses, but stochastically during the transient response or early steady-state for the withdraw responses (*Figure 4A*, right).

To account for the confounding effect of this post hoc observation, we first compared power across the four trial types using an early epoch relative to port entry (250–500 ms). This epoch was chosen with the aim of minimizing the possibility of capturing modulations in the power dynamics by hold-/withdraw-response behavior. There were no significant differences between trial types in this epoch (LMEM ANOVA nonsignificant effect of trial type: $F_{3,749}$ = 0.480, p=0.6935; *Figure 4C*, left), suggesting that the 20–40 Hz transient phase did not differ with response type (hold/withdraw) nor accuracy (correct/incorrect). Although there was a significant effect of trial type when focusing on power during the 250 ms period preceding port withdrawal (LMEM ANOVA: $F_{3,743}$ = 9.451, p<0.001; *Figure 4C*, right), this effect appears primarily due to the post hoc observation mentioned above (as well as the variability observed on InSeq- trials). In fact, InSeq+ trials (hold responses) were higher than OutSeq+ and InSeq- (withdraw responses) trial types ($t_{743}$ = –2.829, p=0.005 and $t_{743}$ = –4.787, p<0.001, respectively), but power in this epoch did not significantly differ between the hold-response trials (InSeq+ and OutSeq-) ($t_{743}$ = –0.458, p=0.647).

To more thoroughly examine whether 20–40 Hz power is associated with accurate temporal order judgments, we used a random forest classifier to test whether power in a set of predefined epochs and across the CA1 axis predicts response accuracy. Given our post hoc observation of fundamentally different dynamics for the hold- vs. withdraw-response trials, all classification analyses focused on comparing accuracy using the hold-response trial types (InSeq+ vs. OutSeq-). This enabled comparisons between correct and incorrect trials while matching motor behavior and nosepoke duration distributions. Classification analysis was first performed using a 16-D feature vector that included 20–40 Hz power during two epochs relative to port entry (200–350, 350–500 ms) and two epochs relative to port withdrawal (–400 to –200, –200 to 0 ms) in each of the four CA1 sites (4 epochs × 4 electrodes) (*Figure 4—figure supplement 1A*, left). We found that these features were not sufficient to discriminate correct and incorrect trials as classifier performance was at chance levels (mean area under curve [AUC], true positive rate [TPR], and false positive rate [FPR] of 0.492, 0.497, and 0.474, respectively) (*Figure 4—figure supplement 1A*, right). As a follow-up, we then used a larger feature space that included time epochs covering the majority of the waveform. We defined a 24-D feature vector that included 20–40 Hz power in four epochs relative to port entry (200–350, 350–500, 500–650 and 650–1200 ms; the latter two not included in the previous analysis) and two epochs relative to port withdrawal (–400 to –200, –200 to 0 ms) in each of the four CA1 sites (6 time epochs × 4 electrodes) (*Figure 4—figure supplement 1B*, left). Using the overlap of lasso (alpha = 0, 0.01, 0.02, and 0.03) and Gini impurity-based relative importance (*Figure 4—figure supplement 1C, D*), the 24-D vector was reduced to three features most predictive of trial accuracy: 20–40 Hz power during the 650–1200 ms period relative to port entry, in the most proximal sites (electrodes 1–3) (*Figure 4—figure supplement 1E*).

We found that these three features were sufficient to discriminate between correct and incorrect temporal order judgments (InSeq+ and OutSeq-, respectively). In fact, the classifier performed with mean areas under the curve (AUC) of 0.588 and 0.477 on the observed and shuffled data, respectively (*Figure 4D*), and a Mann–Whitney *U* test showed a statistically significant difference between the observed and null AUC values (with higher classification performance on the observed data; U = 839,538, p<0.001). Observed compared to null classification receiver operating characteristic (ROC) curve is displayed in *Figure 4D*, with mean observed classifier AUC, TPR, and FPR of 0.588, 0.562, and 0.444, respectively. It is important to note that this prediction analysis uses pooled data across animals.

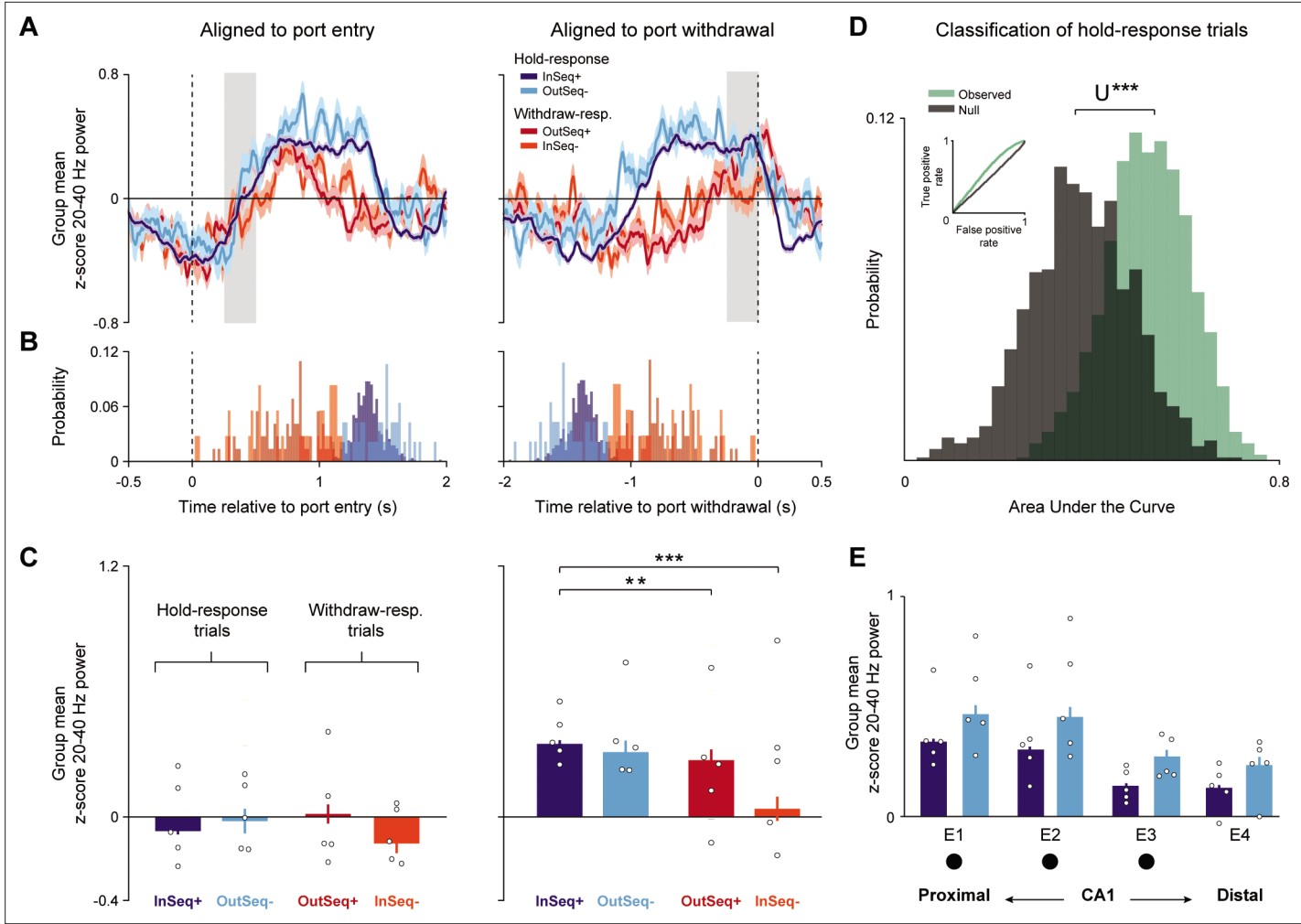

**Figure 4.** CA1 20–40 Hz power steady-state dynamics are attained on hold-response trials and their level predicts trial accuracy. (**A**) Mean 20–40 Hz power traces (± SEM) for the four trial types, with activity aligned to port entry (left) and port withdrawal (right). Trial types are color coded by response type: hold responses are displayed in blue hues (InSeq+, OutSeq-) and withdraw responses in red hues (OutSeq+, InSeq-). Data are from most proximal CA1 electrode during the well-trained session (n = 5 rats). (**B**) Distribution of port withdrawal times (left; relative to port entry) and port entry times (right; relative to port withdrawal) for the trial data shown in (**A**). Note that poke durations are longer for hold-response trials (InSeq+, OutSeq-) and that 20–40 Hz power reached steady-state with longer trial durations. (**C**) Mean z-score 20–40 Hz power in the 250–500 ms period after port entry (left) and in the last 250 ms period prior to port withdrawal (right) for each trial type (epochs indicated by shaded regions in **A**). No differences were observed across conditions in the epoch relative to port entry (nonsignificant effect of trial type: $F_{3,749} = 0.4804$, p=0.6935). Power differed across conditions in the 250 ms period relative to port withdrawal (significant effect of trial type: $F_{3,743} = 9.451$, p<0.001), with significantly lower power during the OutSeq+ and InSeq- (withdraw) trials compared to InSeq+ (hold) trials ($t_{743} = -2.829$, p=0.0048 and $t_{743} = -4.787$, p<0.001, respectively). (**D**) Random forest classifier mean area under the curve (AUC) distributions for classifying between the hold-response trial types (class 1: InSeq+, class 2: OutSeq-) using observed and null data (mean observed and null AUCs of 0.5879 and 0.4772, respectively). Classifier yielded significantly higher performance when using the observed compared to the null data (U = 839538, p<0.001). 3-D classification feature vector consisted of 20–40 Hz power in electrodes 1–3 during the 650–1200 ms period after port entry (see *Figure 4—figure supplement 1A* for feature selection procedure). Inset shows mean observed and null receiver operating characteristics (ROCs) for the classification in (**D**). Classifier mean observed AUC, true positive rate [TPR], and false positive rate [FPR] were 0.5879, 0.5619, and 0.4444, respectively. (**E**) Mean z-score 20–40 Hz power in the 650–1200 ms period (identified through feature selection) in each of the four CA1 sites (filled black circles indicate electrodes identified by feature selection). Steady-state power was significantly higher in the OutSeq- compared to the InSeq+ condition (significant effect of trial type: $F_{1,2406} = 32.687$, p<0.001). Overall power was significantly higher toward proximal CA1 (significant effect of electrode: $F_{3,2406} = 90.573$, p<0.001). Bar plots display mean ± SEM, with individual subject means overlaid as circles. **p<0.01, ***p<0.001 two-tailed *t*-tests (Bonferroni-corrected). U***p<0.001 Mann–Whitney *U* test.

The online version of this article includes the following figure supplement(s) for figure 4:

**Figure supplement 1.** Feature definition and selection for random forest classifier analysis comparing 20–40 Hz power between hold-response trials (InSeq+ vs. OutSeq-).

Post hoc group analyses of the identified feature (650–1200 ms epoch) showed a significant effect of trial type (InSeq+ vs. OutSeq-; LMEM ANOVA: $F_{1,2406}$ = 32.687, p<0.001) and electrode ($F_{3,2406}$ = 90.573, p<0.001), but no significant interaction ($F_{3,2406}$ = 0.193, p=0.901) (*Figure 4E*). Interestingly, power was significantly higher on *incorrect* temporal order judgments (OutSeq- power was higher than InSeq+). Across trial types, power was significantly higher in electrode 1 compared to electrodes 3 and 4 ($t_{2406}$ = –12.310, p<0.001 and $t_{2406}$ = –13.025, p<0.001, respectively). Power in electrode 1 was also higher than power in electrode 2; however, this difference was not significant ($t_{2406}$ = –2.215, p=0.027). Altogether, these results demonstrate that 20–40 Hz steady-state power in proximal CA1 is predictive of accuracy for temporal order judgments during hold-response trials, and that incorrect judgments are associated with significantly higher 20–40 Hz power.

### The increase in CA1 20–40 Hz power with session performance is selective to hold-response trials

*Figure 3* shows increased 20–40 Hz power with session performance, and *Figure 4* shows that 20–40 Hz dynamics that predict trial accuracy occur late in the trial. In linking these results, we aimed to delineate between two possibilities. The first possibility is that increased power with session performance reflects a direct index of sequence knowledge (i.e., CA1 representation of sequence knowledge or retrieval thereof). This would be supported if increased power with performance were selective to

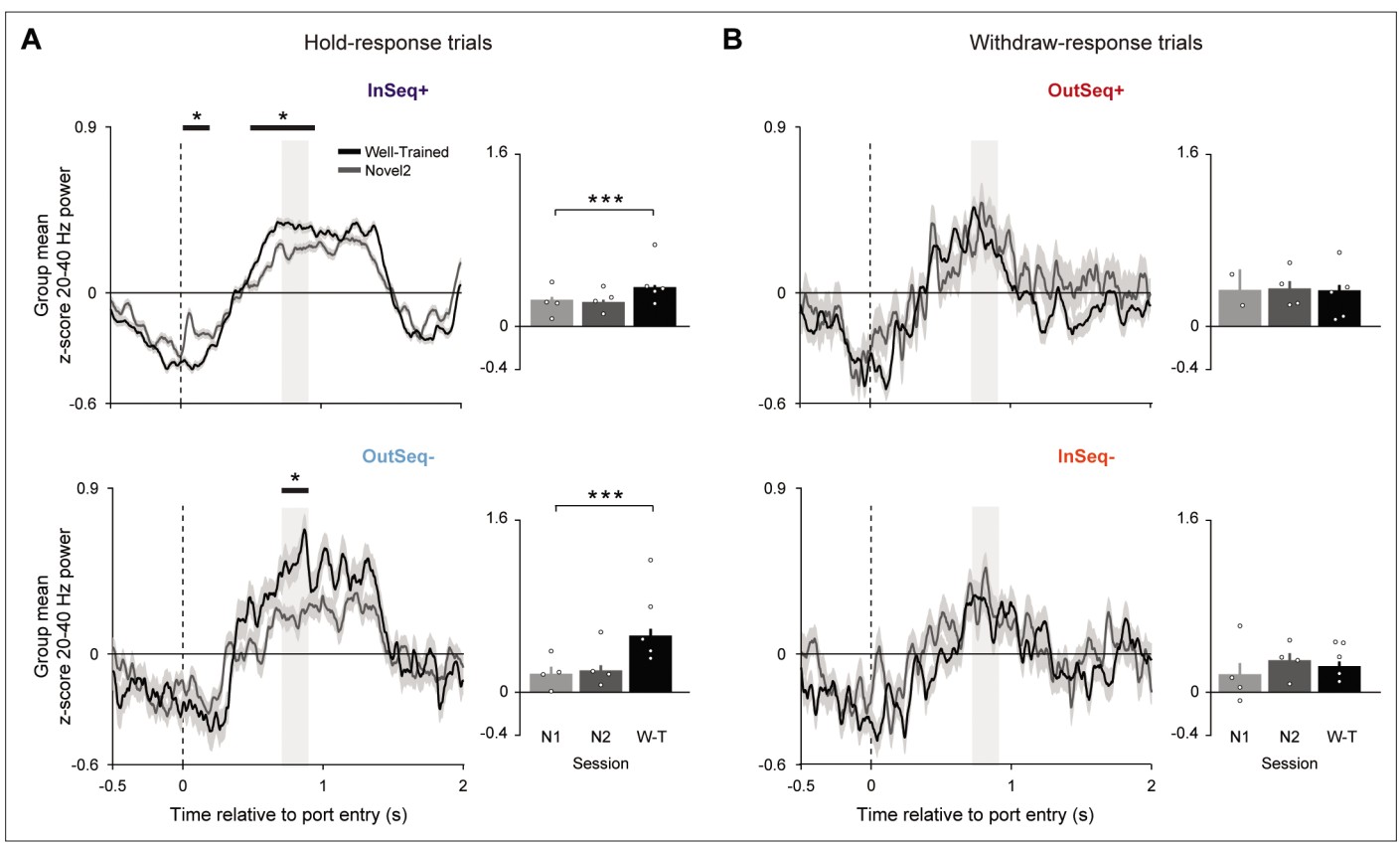

**Figure 5.** Hold-response trials, but not withdraw-response trials, show an increase in 20–40 Hz power with session performance. (**A**) Mean z-score 20–40 Hz power traces (± SEM) for the hold-response trial types (top: InSeq+; bottom: OutSeq-) during the well-trained (n = 5 rats) and novel 2 (n = 4) sessions using the most proximal CA1 site. Significant epochs are indicated with black horizontal bands, performed separately for each trial type (cluster-based permutation testing [CBPT], p<0.05). Bar plots show power (mean ± SEM; individual subject means overlaid as circles) using a fixed epoch across all trial types (significant OutSeq- epoch, the overlap between hold response identified epochs; gray shaded region in line plots). Power during the InSeq+ and OutSeq- trial types significantly increased with session performance, with higher power during the well-trained compared to the novel 1 session (significant effects of performance: $F_{2,1229}$ = 14.184, p<0.001 and $F_{2,173}$ = 12.072, p<0.001, respectively). (**B**) Same as in (**A**), but for the withdraw-response trial types. Power during the OutSeq+ (top) and InSeq- (bottom) trial types did not significantly differ with performance (nonsignificant effects of performance: $F_{2,128}$ = 0.3839, p=0.682 and $F_{2,116}$ = 0.7962, p=0.4534). Individual animal values are overlaid for each bar plot. *Significant epochs identified in CBPT analysis (corrected for multiple comparisons). ***p<0.001 two-tailed t-tests (Bonferroni-corrected).

correct trial types (InSeq+ and OutSeq+). The second possibility is that, given that accuracy predictive dynamics occurred later in the trial (after memory for order has been retrieved and a post-retrieval decision made), power increases reflect a post-decision state that varies with session performance. This would be supported if the session performance effect were selective to trials reaching a post-decision state (hold responses, InSeq+, OutSeq-).

To test these two possibilities, we investigated differences in dynamics across sessions separately for the four trial types (*Figure 5*). Consistent with the latter prediction, our cluster-based permutation testing (CBPT) identified epochs when 20–40 Hz power significantly increased with performance level (excluding data from novel 1 session for sampling reasons). This performance effect was observed for the hold-response trial types (InSeq+, OutSeq-; *Figure 5A*) but not the withdraw-response trial types (*Figure 5B*; which included OutSeq+ trials where order memory retrieval is presumably intact and improved with training). To directly compare power in a fixed epoch across all trial types, the OutSeq- epoch was used since it represented the overlap in significant epochs between the two hold-response trial types. During this epoch, a significant effect of performance level was observed for both hold-response trial types (InSeq+ and OutSeq-; LMEM ANOVA significant effect of performance: $F_{2,1229} = 14.184$, $p<0.001$ and $F_{2,173} = 12.072$, $p<0.001$, respectively). For both trial types, epoch power during the well-trained session was significantly higher than that of the novel 1 session ($t_{1229} = 3.984$, $p<0.001$ and $t_{173} = 4.462$, $p<0.001$, respectively). Lastly, for both trial types, no significant differences in epoch power between novel 1 and novel 2 sessions were observed ($t_{1229} = –0.438$, $p=0.662$ and $t_{173} = 0.748$, $p=0.455$). In contrast, epoch power during both withdraw-response trial types (OutSeq+ and InSeq-) did not significantly differ with performance (LMEM ANOVA nonsignificant effect of performance: $F_{2,128} = 0.384$, $p=0.682$ and $F_{2,116} = 0.796$, $p=0.453$, respectively). These results support the notion that power increases with session performance may reflect a post-decision state that varies with knowledge of the sequence.

## Discussion

We collected LFP activity from the CA1 hippocampal region as rats performed a sequence memory task to identify the frequency content supporting nonspatial information processing. The data presented here expand upon our previous report of 20–40 Hz power during the odor sequence processing periods of the task (*Allen et al., 2016*) by evaluating the behavioral relevance and anatomical distribution of this rhythm across the CA1 proximodistal axis, as well as providing a direct comparison with another behavioral state (running). First, we demonstrated that running and odor sequence processing epochs were characterized by different spectral features. Running was associated with increased power in theta (4–12 Hz) and >24 Hz frequency ranges, whereas odor sequence processing was associated with increased power in theta (at a lower center frequency than during running) and 20–40 Hz ranges. Second, we showed that in both behavioral states there were distinct gradients of recruited frequency power along the CA1 proximodistal axis. During odor processing, 20–40 Hz power was higher in proximal CA1, whereas theta was higher in distal CA1. During running periods, theta was higher in proximal, whereas >24 Hz power was higher in distal CA1. Third, we observed differential performance-related effects across frequency bands; 20–40 Hz power dynamics were positively associated with performance across sessions, while theta and fast gamma demonstrated a negative association. Fourth, we observed differential power dynamics associated with the task's discriminative response requirements. Both hold-response trials (InSeq+ and OutSeq-) and withdraw-response trials (OutSeq+ and InSeq-) displayed a transient phase where 20–40 Hz power gradually increased at a similar rate. However, dynamics subsequently diverged; 20–40 Hz power declined on withdraw-response (presumed OutSeq decisions) trials while hold-response trials (presumed InSeq decisions) displayed a steady-state period of elevated 20–40 Hz power that remained until the animal withdrew. Fifth, we identified the steady phase of 20–40 Hz power in proximal CA1 as a key component carrying task-critical information. In fact, this component of the signal (observed on hold-response trials) increased with session performance and was predictive of the accuracy of temporal order judgments. Collectively, these results suggest that 20–40 Hz power dynamics in the proximal segment of CA1 are linked to processing the temporal order of nonspatial events and reflect a post-decision state that may serve to maintain and protect trial-specific information until a response can be performed.

The nature of this experimental paradigm led to two potential limitations to consider when interpreting the findings. First, the use of a nonspatial discriminative response (hold/withdraw) prevented

us from directly equating response duration across all trial types and unexpectedly led to different power dynamics between hold- and withdraw-response trials (possibly attributable to differences in cue utilization across conditions, as in *Leventhal et al., 2012*, or merely in trial duration). For that reason, we focused our secondary analyses on the effects of response accuracy to hold-response trials. Second, to ensure adequate performance, the task requires that the number of OutSeq trials be kept relatively low (otherwise it is unclear which sequence is being tested). This results in an uneven number of observations across trial types that could disproportionally influence a subset of our analyses. However, we controlled for this possibility by conducting pooled analyses using an LMEM ANOVA to account for correlations in measures obtained from the same subject and by matching sample sizes in the classification analyses. Overall, we believe that these approaches significantly mitigated the potential influence of these confounding factors on the interpretation of our results.

Our results expand our understanding of CA1 spectral profiles across different behavioral states and conditions. First, prior reports have shown that theta and broadband gamma power are recruited during spatial exploration and that power dynamics vary with behavior (*Trimper et al., 2017*; *Chen et al., 2011*; *Ahmed and Mehta, 2012*). We expand on these observations by demonstrating gradients of theta and gamma power along the CA1 proximodistal axis during running and by revealing that a different spectral profile is observed during nonspatial behavior. Second, previous studies have shown a similar recruitment of 20–40 Hz oscillations in odor-based tasks (*Martin et al., 2007*; *Igarashi et al., 2014*; *Rangel et al., 2016*). Oscillations in a similar range (15–35 Hz) were recorded in the hippocampus and olfactory bulb of rats performing a go/no-go odor discrimination task (*Martin et al., 2007*). Interestingly, that study showed a learning-related increase in oscillatory power in the olfactory bulb as well as in the coherence between signals from bulb and hippocampus. However, oscillatory power in the hippocampus did not change as a function of learning. Similarly, 20–40 Hz oscillations were observed in distal CA1 (dCA1) and lateral entorhinal cortex (LEC) of rats engaged in an odor-place association task (*Igarashi et al., 2014*). In that study, learning-related increases in dCA1-LEC coherence were observed, but oscillatory power in either region did not significantly change with learning. This suggests that although these oscillations are observed in a variety of odor-based tasks, increases in odor familiarity or task performance do not necessarily result in a predictable increase in power. Instead, our findings suggest that 20–40 Hz dynamics are linked to specific task demands and, in this case, reflect a preferential engagement of proximal CA1 in processing temporal order judgments.

Evidence for functional heterogeneity along the CA1 proximodistal axis has been previously reported (*Henriksen et al., 2010*; *Hartzell et al., 2013*; *Nakazawa et al., 2016*; *Ng et al., 2018*). Such heterogeneity is not surprising given the known connectivity gradients of the lateral and medial entorhinal cortices (LEC and MEC; *van Strien et al., 2009*; *Witter et al., 2017*), namely, that proximal and distal CA1 are more anatomically and functionally associated with MEC and LEC, respectively. Consequently, the observation of different proximodistal gradients of oscillatory power across studies may result from differences in task demands, which could promote differential engagement of these entorhinal-CA1 circuits (although other factors, including methodological and analytical approaches, may also contribute to this difference). For instance, *Igarashi et al., 2014* showed that 20–40 Hz power and LEC-CA1 coherence were higher in distal CA1 during performance of an odor-place association task (whereas here we show higher power in proximal CA1). Since performance in their task depends on the correct identification of the specific perceptual features that distinguish one odor from another, the power increase in distal CA1 may reflect a stronger engagement of the LEC-dCA1 component of the circuit (as LEC receives strong olfactory input, including direct projections from the olfactory bulb; *Haberly and Price, 1978*; *Agster and Burwell, 2009*). In contrast, in our experiment, identification of the presented odor alone is insufficient for correct performance – the animal must further identify whether the odor is being presented in the appropriate temporal context. The power increase in proximal CA1 we observed may reflect the fact that this additional temporal requirement preferentially engaged the MEC-pCA1 component of the circuit. Interestingly, although MEC is typically associated with spatial navigation functions, this interpretation is supported by a recent study by *Robinson et al., 2017*, in which they demonstrated that optogenetic inactivation of MEC disrupted temporal coding in CA1, while sparing spatial and object coding. Together with the proximodistal pattern we observed, these results suggest that perhaps the MEC→proximal CA1 microcircuit may be important for the processing of nonspatial temporal information.

Oscillations in the 20–40 Hz range (typically referred to as beta) occur widely across the brain and have been associated with different functions across brain regions (*Engel and Fries, 2010*; *Schmidt et al., 2019*). Of particular interest here is the hypothesis that beta serves a gating function that helps lock in and protect a neural state representing trial-critical information (i.e., maintain the status quo; *Engel and Fries, 2010*; *Weiss and Mueller, 2012*). Such a mechanism would be disruptive during encoding or recall processes but advantageous for maintaining a decision until the corresponding response is made, which is consistent with the 20–40 Hz trial-level dynamics we observed in CA1 (low power early in the odor period, but high power in the late period before the response). More precisely, as OutSeq+ trials required animals to perform an 'immediate withdraw-response' (i.e., not wait until the signal), we can use the timing of those responses to infer when temporal order decisions occurred. During those trials, it was the transient phase of 20–40 Hz power that was consistently observed; any encoding and/or recall, as well as the decision based on such processes, necessarily occurred prior to or early in the induction of 20–40 Hz power. This is in line with observations that beta power is attenuated during encoding (*Lundqvist et al., 2016*; *Zavala et al., 2017*) and complements findings that transcranial magnetic stimulation at the beta frequency impairs encoding (*Hanslmayr et al., 2014*). The perspective that beta helps maintain the status quo would predict that the steady-state 20–40 Hz period observed on hold-response trials reflects the system actively maintaining an ensemble or network state in support of the decision to perform a hold response. Though it is possible the post-decision ensemble state reflects a general signal, such as a postural one or a nonspecific signal of a temporal context match, our current finding showed that this epoch is predictive of accuracy and our previous work showed that specific odor and temporal context information is represented at the single-cell (*Allen et al., 2016*) and ensemble (*Shahbaba et al., 2022*) levels during this late trial period. It is therefore possible that what is 'locked in' during the steady-state 20–40 Hz period is the trial-specific content supporting the animal's judgment that the odor is presented in the correct temporal context.

This potential gating function of beta is also consistent with the performance-related effects we observed at the session level. Here, we showed that 20–40 Hz power increases with session performance (lowest in novel 1, highest in well-trained). High beta during the well-trained session possibly reflects higher levels of gating and effective suppression of distracting processes; this is a high-performing session in which the sequence is already well-learned, minimizing the need for additional knowledge acquisition. In contrast, lower beta power in the novel 1 and 2 sessions, which are associated with a higher demand for encoding, may enable flexible knowledge acquisition processes that would otherwise have been interrupted. Consistent with complementary dynamics of beta and gamma rhythms previously reported (*Howe et al., 2011*; *Lundqvist et al., 2016*), we find an inverse pattern for fast gamma power (and theta) with performance, whereby fast gamma power is higher during acquisition of a new sequence (novel 1–2) and decreases with session performance. This finding highlights the distinction between the functional role of 20–40 Hz and that of the other rhythms.

Interestingly, when considering the relationship between steady-state 20–40 Hz power and performance at both the trial and session levels, we noted a pattern more complex than a simple linear relationship. Whereas 20–40 Hz power increased with session performance (from novel 1 to well-trained), steady-state power in the well-trained session was higher during OutSeq- (incorrect hold response) compared to InSeq+ (correct hold response). Although this result may seem counterintuitive, this effect is compatible with the proposed gating function of beta. Specifically, that beta serves to coordinate integration of exogenous information with endogenous information (*Iversen et al., 2009*; *Bastiaansen et al., 2010*), and that beta power regulates the degree of suppression of seemingly distracting processes (*Lundqvist et al., 2016*; *Schmidt et al., 2019*; *Miller et al., 2018*). Proximal CA1 higher steady-state beta levels during incorrect trials may reflect over-suppression, leading to interference with processes needed for successful performance on the task. In other words, the appropriate level of suppression may be advantageous in attenuating distractions, but too high a level may impede essential processes. Consistent with this are clinical reports demonstrating that increased beta in patients with Parkinson's disease is associated with highly sustained, rigid movements (*Chen et al., 2007*; *Pogosyan et al., 2009*; *Brown, 2007*) and difficulty in switching behavioral paradigm (*Stoffers et al., 2001*); clinically disrupting beta power improves these symptoms, with a side effect of inability to ignore distractions and difficulty in focusing on the task at hand (*Cools et al., 2003*; *Moustafa et al., 2008*). Together, these findings point to graded effects of beta on

performance extending to CA1. Alternatively, beta effects may reflect more of a threshold process: higher steady-state power may represent a point of no return, despite retrieval of knowledge that suggests a change in behavior. This latter possibility is consistent with the observation made by *Leventhal et al., 2012*, in which animals were given a second stop cue following an initial go cue. The likelihood of stopping decreased when the second cue occurred during the beta synchronization event elicited from the first cue, reflecting a 'closed gate' on the behavioral state initially cued. Future studies are needed to adjudicate between these possibilities, possibly by specifying the relationship between the information content represented in the ensemble activity and the 20–40 Hz power dynamics.

Future studies will also be needed to determine the degree to which our findings relate to cue utilization. Utilization of a sensory cue to inform motor output has been shown to recruit ~20 Hz (beta) in cortico-basal ganglia circuits (CBG, *Leventhal et al., 2012*), CA1-striatal circuits (*Lansink et al., 2016*), and the dentate gyrus (DG; *Rangel et al., 2015*), giving rise to a working hypothesis for the functional role of ~20 Hz in sensorimotor integration. Since, in this study, animals utilized the odor cue to either maintain or terminate a hold posture, it is possible that this aspect of sensorimotor integration contributed to the oscillations we observed. Unfortunately, resolving this issue would require a different experimental design. Similarly, further work will be needed to determine the origin of the 20–40 Hz rhythm in CA1. CA1 receives input from a number of other sources, including entorhinal cortex, CA2, CA3, and the medial septum (*van Strien et al., 2009*), and the contributions of these upstream structures to the generation of this oscillation in CA1 remain unknown. Though we previously showed that the spiking activity of a proportion of CA1 neurons is phase-locked to these 20–40 Hz oscillations (*Allen et al., 2016*), we cannot rule out the possibility that volume conduction contributed to the presence of these rhythms and their gradients. A prior study performed current source density analysis for hippocampal 15–35 Hz power, spanning DG, CA3, and CA1, and localized this rhythm to the DG (*Rangel et al., 2015*). Altogether, future experiments are needed to clarify the relative contribution of mnemonic and sensorimotor processes to the observed rhythms and their gradients, as well as the potential influence of volume conduction.

Finally, evidence from multisite recordings suggest the 20–40 Hz dynamics we observed in CA1 reflect task-critical, multiregional integration. As mentioned earlier, coherence in the beta range between the hippocampus and olfactory regions (*Martin and Ravel, 2014*; *Kay, 2014*), as well as with the entorhinal cortex (*Igarashi et al., 2014*), has been previously reported during odor-dependent tasks. Importantly, these dynamics are not restricted to olfactory processing (visual cues: *Lansink et al., 2016*; auditory cues: *Leventhal et al., 2012*) nor do they exclusively coordinate hippocampus with olfactory or medial temporal lobe regions. *Lansink et al., 2016* demonstrated CA1 beta (15–20 Hz) recruitment as rats responded in the presence of a reward-predictive visual cue that directed navigation (as opposed to goal-directed navigation absent a cue). Notably, during this recruitment neuronal firing was synchronized between CA1 and ventral striatum (*Lansink et al., 2016*). Specific to our task, structures connected to the hippocampus have also been shown to be critical for performance, including prefrontal, perirhinal, and mid-thalamic regions (*Fortin et al., 2016*; *Jayachandran et al., 2019*). As each structure likely contains region-specific representations, beta may serve to bind these regions together to coordinate these distinct task-critical representations across regions and support goal-directed behavior (*Fries, 2005*). To clarify, multiregional integration may serve as the general function of beta, which, in our paradigm, is in the service of supporting order memory judgments. Increases in CA1 beta have been reported in other task conditions and demands, such as exploration and novelty (*Berke et al., 2008*; *França et al., 2014*), which could reflect multiregional integration supporting other forms of task-critical information processing. Collectively, these findings provide evidence for the far-reaching role of beta in coordinating information processing across brain regions by gating their region-specific patterns of activity important for the task at hand.

## Materials and methods

Our group previously published using the same dataset, and a detailed description of the methods can be found in *Allen et al., 2016*. The methods are summarized below.

## Subjects

Five male Long–Evans rats were used in this study. Rats were ~350 g and ~3 months of age at the beginning of the experiment. Animals were water restricted for optimum task engagement but were provided full access to water on weekends. Proper hydration levels were monitored throughout the experiment. All procedures were conducted in accordance with the guidelines for care and use of laboratory animals published by the National Institutes of Health. All animals were handled according to an approved Institutional Animal Care and Use Committee (IACUC) protocol (protocol AUP-20-174).

## Equipment

The apparatus used for this task consisted of a linear track with water ports on either end for water reward delivery. One end of the maze contained an odor port (above the water port) connected to an automated odor delivery system. Photobeam sensors detected when the animal's nose entered and withdrew from the odor port. Detection of port entry triggered odor delivery. Separate tubing lines were used for each odor item; however, all converged at a single channel at the bottom of the odor port. The odor port was kept clear of previous odor traces using a negative pressure vacuum located at the top of the port. A 96-channel Multichannel Acquisition Processor (MAP; Plexon) was used to interface the hardware (Plexon timing boards and National Instruments input/output devices) in real time and record the behavioral and electrophysiological data as well as control the hardware.

## Odor sequence task

In this hippocampus-dependent task, rats were presented with a series of five odors delivered one at a time in the same odor port (*Figure 1A*). In each session, the same sequence was presented multiple times, with approximately half the presentations including all items 'in sequence' (InSeq; ABCDE) and the other half including one item 'out of sequence' (OutSeq; e.g., AB*D*DE) (*Figure 1B*). Each odor presentation was initiated upon port entry and rats were required to correctly identify the odor as either InSeq (by holding their nosepoke response until an auditory signal at 1.2 s) or OutSeq (by withdrawing their nose before the signal; <1.2 s) to receive a water reward (*Figure 1C*). The auditory signal indicates to the animal that it had correctly held its nose in the port long enough to receive a reward. The presentation of the sequence terminated upon an incorrect response (e.g., the fifth odor was only presented if the four preceding odors were correctly identified as InSeq or OutSeq). After termination of a sequence, regardless of accuracy, the animal was required to run to the opposite side of the track, then return back to the side with the odor port to begin the next sequence presentation. Animals were trained preoperatively on sequence ABCDE (lemon, rum, anise, vanilla, and banana) until they reached asymptotic performance (>80% correct on both InSeq and OutSeq trials; ~6 weeks). Following surgical recovery, electrophysiological data was collected as animals performed the same sequence (ABCDE), followed by two consecutive sessions using a novel sequence (VWXYZ; almond, cinnamon, coconut, peppermint, and strawberry), referred to as novel 1 and novel 2, respectively (*Figure 1D*). See *Supplementary file 1* for individual animal trial counts in each session.

## Surgery

Rats received a preoperative injection of the analgesic buprenorphine (0.02 mg/kg, s.c.) ~10 min before induction of anesthesia. General anesthesia was induced using isoflurane (induction: 4%; maintenance: 1–2%) mixed with oxygen (800 ml/min). After being placed in the stereotaxic apparatus, rats were administered glycopyrrolate (0.5 mg/kg, s.c.) to help prevent respiratory difficulties. A protective ophthalmic ointment was then applied to their eyes, and their scalp was locally anesthetized with marcaine (7.5 mg/ml, 0.5 ml, s.c.). Body temperature was monitored and maintained throughout surgery, and a Ringer's solution with 5% dextrose was periodically administered to maintain hydration (total volume of 5 ml, s.c.). The skull was exposed following a midline incision, and adjustments were made to ensure the skull was level. Six support screws (four titanium, two stainless steel) and a ground screw (stainless steel; positioned over the cerebellum) were anchored to the skull. A piece of skull ~3 mm in diameter (centered on coordinates: –4.0 mm anteroposterior, 3.5 mm mediolateral) was removed over the left hippocampus. Quickly after the dura was carefully removed, the base of the microdrive was lowered onto the exposed cortex, the cavity was filled with Kwik-Sil (World Precision Instruments), the ground wire was connected, and the microdrive was secured to the support skull screws with dental cement. Each tetrode was then advanced ~900 μm into the brain. Finally,

the incision was sutured and dressed with Neosporin and rats were returned to a clean cage, where they were monitored until they awoke from anesthesia. One day following surgery, rats were given an analgesic (flunixin, 2.5 mg/kg, s.c.) and Neosporin was reapplied to the incision site.

## Electrophysiological recordings

Both spiking and LFP activity were recorded from the CA1 pyramidal layer of the dorsal hippocampus as rats performed the task (see *Allen et al., 2016*), but this study focuses exclusively on a detailed analysis of the LFP activity. Each chronically implanted microdrive contained 20 independently drivable tetrodes, with each tetrode consisting of four twisted nichrome wires (13 µm in diameter; California Fine Wire) gold-plated to achieve a final tip impedance of ~250 kΩ (measured at 1 kHz). Following the surgical recovery period, tetrodes were slowly advanced over a period of ~3 weeks while monitoring electrophysiological signatures of the CA1 pyramidal cell layer (e.g., sharp waves, ripples, and theta amplitude). Voltage signals from electrode tips were referenced to a ground screw positioned over the cerebellum. LFP activity was filtered (1.5–400 Hz), amplified (1000×), digitized (1 kHz), and recorded to disk with the data acquisition system (MAP, Plexon). Neural activity was first recorded on the odor sequence learned before surgery (ABCDE; 'well-trained' session), followed by two consecutive sessions on the same novel sequence (VWXYZ; novel 1 and novel 2 sessions). At the end of the experiment, recording sites were confirmed by passing current through the electrodes (10 µA for 10 s; positive to electrode, negative to implant ground) before perfusion (0.9% PBS followed by 4% paraformaldehyde) to produce small marking lesions, which were subsequently localized on Nissl-stained tissue slices (cresyl violet staining).

## Selection of electrodes along the proximodistal axis

In order to sample four representative electrodes along the proximodistal axis of CA1, we chose the first and the last electrodes (most proximal and most distal, respectively) and two electrodes in between that were equidistant. We confirmed the relative spatial distribution of these electrodes as well as their localization within the pyramidal layer of CA1 based on standard spectral properties during baseline, odor sampling, and running periods. For each of the four electrodes selected per animal, LFP activity patterns were confirmed in adjacent electrodes (from the remaining subset of 16 electrodes).

## Preprocessing and spectral analysis

The raw data was preprocessed using a Butterworth notch filter to remove 60 Hz line noise. Artifact rejection was defined by time indices with voltage values 5 standard deviations away from the mean of the entire recording in the same channel. Artifact time points were included for the wavelet convolution in order to maintain the temporal structure of the data, but their associated power values were removed before computing the mean and standard deviation for baseline normalization (see below). Any trial containing an artifact was excluded from analyses. For spectral analysis, we utilized the Wavelet toolbox in MATLAB (MathWorks) to generate analytic Morlet wavelets for frequencies between 3 and 250 Hz. These wavelets were tested and verified on simulated data with known spectral properties. Next, we extracted behavior-locked instantaneous power in the specified frequency ranges. In all analyses, the first odor of each sequence was excluded as it was always preceded by running, whereas the animal was stationary prior to all other odor positions.

## Normalization

Instantaneous power is reported as a z-score value relative to the mean and standard deviation of power for a given frequency calculated from a 30 min subset of the recording from the same electrode. For comparison, we also used two additional normalization approaches. One approach calculated z-scores relative to the other time points within the same trial (0–1.5 s for trials aligned to port entry; –1 s to 0 s for trials aligned to port withdrawal). In the second approach, power for a given time point and frequency within a trial was divided by the sum of the power across all trial time points in the same frequency, which captured percentage increase in power at a given frequency. As all three methods yielded comparable results, the reported results relied on the z-normalization to the 30 min recording subset. As this 30 min period included a variety of behavioral and cognitive states, including

odor sampling, running, grooming, and reward consumption, it offers a better characterization of the variance of spectral dynamics associated with the animal's experiences.

## Sample sizes, replicates, and exclusion criteria

Sample sizes were determined using standards in behavioral electrophysiology experiments. Data was recorded from five animals, with each animal providing data from 20 electrodes (tetrodes) and several trials per session (range: 75–219; see *Supplementary file 1*). Although it takes several months to train, implant, and record from each animal, the 'experiment' focused on three daily sessions (well-trained, novel 1, and novel 2 sessions) matched across animals. Data from the same animal was *not* collapsed across sessions. In our design, we view animals as biological replicates and, within each animal, the number of trials as technical replicates. Electrodes can be viewed as biological replicates since we compared effects as a function of electrode position within the CA1. Standard preprocessing approaches were used to exclude data contaminated by electrical noise or artifacts. Trials with artifacts associated with bumping or touching the headstage (voltage values >5 SD above or below the mean) were automatically excluded. Note that this exclusion was performed before (and blind to) analysis of the results. No additional statistical outliers were removed. *Supplementary file 1* provides detailed information on the number of trials included for each animal and each statistical comparison.

## Linear mixed-effects model (LMEM) ANOVA

Data were first tested for normality using the Anderson–Darling test in MATLAB (adtest function). All data met the normality assumption, except for the classification performance output measures (nonparametric tests were performed on the latter). To incorporate all observations in the ANOVAs, while accounting for correlations in scores from the same subjects, an LMEM was then used for all group analyses unless otherwise stated. More specifically, the LMEM, which generalizes the conventional general linear model, includes both fixed effects (e.g., electrodes, trials, sessions) and random effects (subjects). Corresponding coefficients and intercepts were estimated for both types of effects, such that the fixed effects reported exclude the variation due to intra-subject correlations. LMEM fitting and marginal ANOVAs were done using the fitlme and anova MATLAB functions, respectively. The restricted maximum likelihood method was used for model parameter estimation and the Cholesky parametrization for estimation of the random effects term covariance matrix. It is important to note that standard ANOVAs, either using one value per subject or collapsing observations across subjects, produced a comparable pattern of results, though the LMEM approach is more statistically rigorous. The Bonferroni correction was applied to all subsequent pairwise comparisons.

## Cluster-based permutation testing (CBPT) analyses

CBPT was used to agnostically identify group-level significant time epochs that differed with session performance as a function of the full power time course (*Figures 3B and 5*). Briefly, this involved calculating a t-statistic for each time point, between the novel 2 (insufficient correct trial counts to perform this contrast using novel 1) and well-trained conditions, generating an observed t-vector. A null distribution of t-vectors was also generated over 1000 unpaired permutations, whereby in each permutation, condition labels were shuffled. A p-value for each time point was obtained by comparing the observed to the null distribution of t-values at the same time point, thereby generating a p-vector. To correct for multiple comparisons, clusters of contiguous time points with a p<0.05 were identified and compared to the null distribution cluster size. Observed clusters with sizes larger than the 95th percentile value of null distribution cluster size were considered significant (means of correction for multiple comparisons).

## Random forest classifier analyses

Classification analyses were performed to compare 20–40 Hz power levels between InSeq+ and OutSeq- trials; these two trial types were matched for response type (hold) but differed in response accuracy (correct/incorrect, +/-). Support vector machine (SVM), logistic regression, and random forest were three preselected classifiers that were trained and tested using leave-one-out cross-validation, implemented using scikit-learn in Python. For all classification analyses, trial number was fixed between the two classes. Since class 2 (OutSeq-) contained a smaller number of trials compared to class 1 (InSeq+), 1000 iterations were performed where, in each iteration, classifier performance

was computed using a random subsample of class 1 equal in size to class 2. This yielded an observed distribution of AUC, and TPR and FPR performance values. For the null distribution, 1000 permutations were done whereby, in each permutation, class 1 and 2 observations were randomly shuffled, yielding an arbitrary split of two classes. Classifier performance was calculated for each permutation, yielding null distributions for AUC, TPR, and FPR performance measures. Lastly, since observed distributions of classifier performance measures were not normally distributed, a Mann–Whitney *U* test was used to test for differences between the null and observed AUC distributions. Random forest showed the best performance and is therefore the focus of the classifier results reported here.

Random forest classification performance was tested using the predefined time windows (*Figures 3, 4A and C*) with a 16-D feature vector including 20–40 Hz power in four time windows (200–350 and 350–500 ms relative to port entry; –400 to –200 and –200 to 0 ms relative to port withdrawal) windows and the four CA1 sites (*Figure 4—figure supplement 1A*). This enabled the investigation of whether the joint values of these features are sufficient to classify performance – an inference that is not extracted from *Figure 4C*. Then, implementation was done using a 24-D feature vector defined as 20–40 Hz power in four epochs relative to port entry (200–350, 350–500, 500–650, 650–1200 ms) and two epochs relative to port withdrawal (–400 to –200, –200 to 0 ms), in each of the four CA1 sites (6 time epochs × 4 CA1 sites) (*Figure 4—figure supplement 1B*). Lasso and Gini impurity were subsequently used to further reduce the dimension of the starting feature vector with the aim of identifying which of the 24 features contributed to classification accuracy. Using lasso feature selection, sparsity was promoted over a range of alphas (0, 0.01, 0.02, and 0.03) (*Figure 4—figure supplement 1C*). Features with nonzero weights across all alphas were compared with the five features with highest relative importance using Gini impurity (*Figure 4—figure supplement 1D*). The overlap between lasso feature selection and Gini relative importance consisted of three features (20–40 Hz power in 650–1200 ms period after port entry in the three most pCA1 electrodes; *Figure 4—figure supplement 1E*). These three features were utilized for the final classification analysis presented in *Figure 4D and E*.

## Acknowledgements

This research was supported in part by the National Science Foundation (awards IOS-1150292 and BCS 1439267 to NJF), the National Institutes of Health (awards R01 MH115697 and R01 DC017687 to NJF; R01 MH102392 and R01 AG053555 to MAY; T32 NS45540 support for SG; and T32 DC010775 support for GAE), and the Whitehall Foundation (award 2010-05-84 to NJF). We also thank Javad Karimi Abadchi, Amy Daitch, Aaron Gudmundson, and members of the Fortin Lab for useful discussions on the present work.

---

## Additional information

### Funding

| Funder | Grant reference number | Author |
| --- | --- | --- |
| National Science Foundation | CAREER Award IOS-1150292 | Norbert J Fortin |
| National Science Foundation | BCS 1439267 | Norbert J Fortin |
| National Institutes of Health | R01 MH115697 | Norbert J Fortin |
| National Institutes of Health | R01 DC017687 | Norbert J Fortin |
| National Institutes of Health | R01 MH102392 | Michael A Yassa |
| National Institutes of Health | R01 AG053555 | Michael A Yassa |

| Funder | Grant reference number | Author |
|--------|------------------------|--------|
| National Institutes of Health | Training Grant T32 NS45540 | Sandra Gattas |
| National Institutes of Health | Training Grant T32 DC010775 | Gabriel A Elias |
| Whitehall Foundation | 2010-05-84 | Norbert J Fortin |

The funders had no role in study design, data collection and interpretation, or the decision to submit the work for publication.

## Author contributions

Sandra Gattas, Conceptualization, Data curation, Formal analysis, Investigation, Validation, Visualization, Writing - original draft, Writing – review and editing; Gabriel A Elias, Data curation, Investigation, Writing – review and editing; John Janecek, Investigation, Methodology; Michael A Yassa, Funding acquisition, Investigation, Supervision, Writing – review and editing; Norbert J Fortin, Conceptualization, Data curation, Funding acquisition, Investigation, Methodology, Project administration, Supervision, Writing – review and editing

## Author ORCIDs

Sandra Gattas http://orcid.org/0000-0003-1608-1469
Gabriel A Elias http://orcid.org/0000-0002-2339-7245
Michael A Yassa http://orcid.org/0000-0002-8635-1498
Norbert J Fortin http://orcid.org/0000-0002-6793-6984

## Ethics

All procedures were conducted in accordance with the guidelines from care and use of laboratory animals published by the National Institutes of Health. All animals were handled according to an approved Institutional Animal Care and Use Committee (IACUC) protocol (Protocol AUP-20-174).

## Decision letter and Author response

Decision letter https://doi.org/10.7554/eLife.55528.sa1
Author response https://doi.org/10.7554/eLife.55528.sa2

## Additional files

### Supplementary files

• Supplementary file 1. Trial counts for each session and animal. Values in parentheses indicate trials included in the analyses (after artifact rejection; counts from most proximal electrode).

• Transparent reporting form

### Data availability

Data available on Dryad, Data DOI: https://doi.org/10.7280/D11960.

The following dataset was generated:

| Author(s) | Year | Dataset title | Dataset URL | Database and Identifier |
|-----------|------|---------------|-------------|-------------------------|
| Gattas S, Elias G, Janecek J, Yassa MA, Fortin NJ | 2022 | Proximal CA1 20-40 Hz power dynamics reflect trial-specific information processing supporting nonspatial sequence memory | https://dx.doi.org/10.7280/dryad.D11960 | Dryad Digital Repository, 10.7280/dryad.D11960 |

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
