## [Editor Report]

This article presents intriguing evidence that 20–40 Hz amplitude increases in the hippocampus are tied to task-relevant parameters, namely, odors presented in a sequence, as well as learning. The results reveal new insights about hippocampal processing of nonspatial information and contribute to a greater understanding of hippocampal network mechanisms of memory processing.

---

## [Decision Letter]

**Decision letter after peer review:**

Thank you for submitting your article "CA1 20-40 Hz oscillatory dynamics reflect trial-specific information processing supporting nonspatial sequence memory" for consideration by *eLife*. Your article has been reviewed by 2 peer reviewers, and the evaluation has been overseen by a Reviewing Editor and Laura Colgin as the Senior Editor. The following individual involved in review of your submission has agreed to reveal their identity: Diego Restrepo (Reviewer #2).

The reviewers have discussed the reviews with one another and the Reviewing Editor has drafted this decision to help you prepare a revised submission.

The editors and reviewers have judged that your manuscript is of interest, but as described below that additional analyses are required to support the conclusions. Because of this, we cannot guarantee ultimate acceptance of your manuscript at this time, as it depends on the results of the additional analyses.

Summary:

In this manuscript, the authors characterize behavioral parameters associated with 20-40 Hz amplitude increases in the CA1 regions of the hippocampus. They present intriguing evidence that 20-40Hz amplitude increases are tied to task-relevant parameters (odors presented in a sequence), learning, specific regions of CA1 (proximal as opposed to distal CA1). The authors propose that the 20-40Hz oscillations are associated with processing of nonspatial information, which may be critical for judgments that depend upon order memory. Although the findings were found to be of interest, the results as presented were viewed as insufficient to support the authors' claims.

Essential revisions:

1. It is unclear that the oscillatory dynamics presented are tied specifically to nonspatial information or ordering, as opposed to expectancy, the maintenance of an active assembly, or utilized cues more generally (as has been shown in several other brain regions during similar tasks). A number of additional analyses would be required to support these arguments (depending upon the outcome). A number of analyses are suggested below:

A) Related to "accurately performing the cognitive operations": The authors pool correct (InSeq+ and Outseq+) trials and compare them to incorrect (Inseq- and Outseq-) trials to assess oscillatory differences associated with accuracy. It seems that the following additional comparisons should be made: InSeq+ and InSeq- (controls for stimulus and trial type), Outseq+ and Outseq- (controls for stimulus and trial type). Of note, the differences that the authors observe in the comparison they performed could be relevant to the Leventhal et al., 2009 study listed below. Specifically, they observe differences in β amplitude for successfully utilized go and no-go cues, and smaller amplitude oscillations for unsuccessfully utilized cues. The authors should address this in their discussion because it provides an interpretation that is not specific to odors or order memory.

B) Related to the processing of "nonspatial information critical for order memory judgements": It is unclear that the authors are examining links to order as opposed to general expectancy, and it is not readily clear how they might tease this apart (as opposed to pull back their claims a bit). An interesting analysis could examine responses to progressive odors in a sequence. How do Outseq+ oscillatory dynamics for the third out of sequence odor differ from a) the first order of a sequence (high expectancy, perhaps not really part of a sequence as the first odor), b) the second odor in the Outseq+ sequence (high expectancy, consistent order), b) the fourth odor in an Outseq+ sequence (which is the same in InSeq and OutSeq trials, and therefore possibly high expectancy, but an order violation). The authors could also consider analyses similar to those used in Bastiaansen et al., 2010. That study reported a progressive increase in β oscillation amplitude with successive words in congruent sentences, but a drop in β oscillations at the point of an incongruent word in a sentence.

2. The authors need to integrate their findings with the large existing body of work that has been done investigating 20-40Hz β/low γ oscillations in the hippocampus and other brain regions during similar tasks. The finding currently feels isolated from previous work conducted in this realm, and integrating their findings with the existing literature would go a long way towards placing their findings in an existing framework and improving the impact of their work. The authors should read and cite the following studies at minimum (note that some are reviews that summarize many studies as a starting point, with a few particularly relevant original research papers listed below), integrating their results with previously published data and addressing leading theories regarding mechanisms and proposed functions. Knowledge of the below studies will provide a larger framework for their study, substantially increase the impact, and aid in the interpretation of results (particularly the few that also address 20-40Hz oscillations in the hippocampus).

Martin, Claire, Jennifer Beshel, and Leslie M Kay. 2007. "An Olfacto-Hippocampal Network Is Dynamically Involved in Odor-Discrimination Learning." Journal of Neurophysiology 98 (4): 2196-2205.

Berke, Joshua D, Vaughn Hetrick, Jason Breck, and Robert W Greene. 2008. "Transient 23-30 Hz Oscillations in Mouse Hippocampus during Exploration of Novel Environments." Hippocampus 18 (5): 519-29.

Iversen, John R, Bruno H Repp, and Aniruddh D Patel. 2009. "Top-Down Control of Rhythm Perception Modulates Early Auditory Responses."

Bastiaansen, Marcel, Lilla Magyari, and Peter Hagoort. 2010. "Syntactic Unification Operations Are Reflected in Oscillatory Dynamics during On-Line Sentence Comprehension." Journal of Cognitive Neuroscience 22 (7): 1333-47.

Weiss, Sabine, and Horst M Mueller. 2012. "'Too Many Betas Do Not Spoil the Broth': The Role of Β Brain Oscillations in Language Processing." Frontiers in Psychology 3 (January): 201.

Leventhal, Daniel K, Gregory J Gage, Robert Schmidt, Jeffrey R Pettibone, Alaina C Case, and Joshua D Berke. 2012. "Basal Ganglia Β Oscillations Accompany Cue Utilization." Neuron 73 (3): 523-36.

Kay, Leslie M. 2014. "Circuit Oscillations in Odor Perception and Memory." Progress in Brain Research 208 (January): 223-51.

Rangel, Lara M., Jon W Rueckemann, Pamela D. Riviere, Katherine R. Keefe, Blake S. Porter, Ian S. Heimbuch, Carl H. C.H. Budlong, and Howard Eichenbaum. 2016. "Rhythmic Coordination of Hippocampal Neurons during Associative Memory Processing." *eLife* 5 January 2016.

Trimper, John B., Claire R. Galloway, Andrew C. Jones, Kaavya Mandi, and Joseph R. Manns. 2017. "Γ Oscillations in Rat Hippocampal Subregions Dentate Gyrus, CA3, CA1, and Subiculum Underlie Associative Memory Encoding." Cell Reports 21 (9): 2419-32.

3. The authors averaged across four electrodes (spanning the proximal distal axis) for the analyses in Figure 4. It is unclear why the analysis in the latter figures was performed with an average for all electrodes across the proximodistal axis. Figures 1 and 2 show clear differences along the proximodistal axis. For example, in Figure 2 there appear to be significant changes in power as the animal learns in the most proximal electrode for theta and high γ, but this is not mentioned in the manuscript. It is unclear that this averaging is appropriate given the differences across the electrodes. The authors should perform these comparisons separately for proximal and distal electrodes at least. Importantly, if the authors analyze power changes in each location, are there changes in power that were missed by averaging across electrodes? Are there differences in oscillations between ABCDE and VWXYZ?

4. The authors were very straightforward about their trial numbers in their supplementary data. It is unclear that there are enough trials for some of the comparisons. This is perhaps why some of the data was pooled in the manner shown in Figure 4. However, it is interesting to note that trends for individual rats do not necessarily follow the average. On that note, are parametric statistics appropriate for all comparisons?

5. It is unclear why the authors analyze data aligned to the end of the trial for several of their analyses (as opposed to the beginning of a trial). There is an assumption embedded in the decision that interesting oscillatory dynamics in both InSeq and Outseq trial types occur before the end of a poke period (as opposed to a decision point earlier). The authors could be missing important dynamics/differences at the beginning InSeq odor presentations, and should consider the alignment to the beginning of odor presentation as well. Perhaps a thorough analysis would report differences across running windows along the epoch (which would strengthen their argument that the differences are greater towards the middle or end of an interval).

6. The authors indicate in multiple places that increases in the amplitude of 20-40Hz oscillations increase over the duration of the odor presentation/poke maintenance period. This is an interesting finding by itself and the authors should quantify this with statistics. A similar finding was reported previously in Rangel et al., 2016.

7. A major problem with the presentation of the results is that there is no introduction to the task and the recordings. This is particularly important for the broad readership in *eLife*. As it is, the reader has to go to the methods where the explanation of the task is unclear. Furthermore, Figure 1 is not very informative and should be modified to convey information on the task, the fact that the time in the port is difference for InSeq and OutSeq, and the classification of trials as InSeq+, InSeq-, OutSeq+ and OutSeq-. Also, please provide an example of raw data recorded from one electrode and the Morlet wavelet analysis for that electrode. One would expect that the wavelet analysis would show clearly the theta phase aligned β (and γ?) bursts that become smoothed in the average data shown throughout the manuscript.

8. Presentation of the behavioral outcome is lacking. How well did the animals perform in "lowest performance", "moderate performance" and "highest performance" in Figure 2? Please provide the data quantifying performance. Similarly, are the data in Table S4 statistically significant (as opposed to random)? Please show data for ABCDE and VWXYZ in the table.

9. The validity of the analysis of the data in Figure 4 showing that power varies across trial types is questionable. The authors had to merge and subsample the data. Did the authors attempt to decode trial types using machine learning or discriminant analysis? Perhaps this would provide more reliable information. Decoding analysis would allow calculation of decoding accuracy throughout the time course alleviating the problem with comparisons when the animal has left the port. Did the authors attempt to do this analysis using electrode locations that provided the largest differences in oscillatory power for the different trial outcomes?

10. In addition to p-values, please also consistently provide the statistic, degrees of freedom, and make sure that the number of animals tested is consistently clear.

---

## [Author Response]

Essential revisions:1. It is unclear that the oscillatory dynamics presented are tied specifically to nonspatial information or ordering, as opposed to expectancy, the maintenance of an active assembly, or utilized cues more generally (as has been shown in several other brain regions during similar tasks). A number of additional analyses would be required to support these arguments (depending upon the outcome). A number of analyses are suggested below:A) Related to "accurately performing the cognitive operations": The authors pool correct (InSeq+ and Outseq+) trials and compare them to incorrect (Inseq- and Outseq-) trials to assess oscillatory differences associated with accuracy. It seems that the following additional comparisons should be made: InSeq+ and InSeq- (controls for stimulus and trial type), Outseq+ and Outseq- (controls for stimulus and trial type).

As detailed in our response to point #5 below, additional analyses have revealed a relationship between β power and hold duration (β gradually increases within odor presentations and reaches a steady-state after ~500ms). This confounds many of the trial-type comparisons, as some involve comparing trials of different durations (e.g., InSeq+ vs InSeq-) and others early epochs where power is low (e.g., InSeq- vs OutSeq+). For that reason, we now focus on the accuracy effect by comparing InSeq+ vs OutSeq- trials, which have comparable (long) response durations but differ in accuracy. Nevertheless, for completeness, data from the four trial types (InSeq+, OutSeq+, OutSeq-, InSeq-), aligned to both port entry and port withdrawal, is now shown in Figure 4C.

Of note, the differences that the authors observe in the comparison they performed could be relevant to the Leventhal et al., 2009 study listed below. Specifically, they observe differences in β amplitude for successfully utilized go and no-go cues, and smaller amplitude oscillations for unsuccessfully utilized cues. The authors should address this in their discussion because it provides an interpretation that is not specific to odors or order memory.

We thank the reviewers for pointing us to the work by Leventhal and colleagues. We have modified our analyses in Figure 4 (added analyses aligned to port entry), performed a new analysis showing that the session performance effect from Figure 3 is specific to hold-response trials (Figure 5), and added the corresponding interpretations in the discussion (p15). Overall, our results are consistent with the notion that cue-utilization recruits β dynamics in CA1.

B) Related to the processing of "nonspatial information critical for order memory judgements": It is unclear that the authors are examining links to order as opposed to general expectancy, and it is not readily clear how they might tease this apart (as opposed to pull back their claims a bit). An interesting analysis could examine responses to progressive odors in a sequence. How do Outseq+ oscillatory dynamics for the third out of sequence odor differ from a) the first order of a sequence (high expectancy, perhaps not really part of a sequence as the first odor), b) the second odor in the Outseq+ sequence (high expectancy, consistent order), b) the fourth odor in an Outseq+ sequence (which is the same in InSeq and OutSeq trials, and therefore possibly high expectancy, but an order violation). The authors could also consider analyses similar to those used in Bastiaansen et al., 2010. That study reported a progressive increase in β oscillation amplitude with successive words in congruent sentences, but a drop in β oscillations at the point of an incongruent word in a sentence.

Thank you for bringing this up – we now realize that we were not sufficiently clear on this. To clarify, as mentioned in the title, our main finding is that we show CA1 β dynamics that reflect “trial-specific information processing supporting nonspatial sequence memory.” More specifically, our results suggest CA1 β increases reflect a post-decision state that may serve to protect local information processing from disruption until a response can be performed. In our paradigm, the trial-specific information processing is in the support of order decisions (the key task demand). We are not arguing that this role of β is limited to order memory – it could support another form of task-critical information processing in a different task. This has been clarified in the introduction (last paragraph), results (p11-12), and discussion (p13-14).

Also, thank you for recommending analyses to examine the effect of general expectancy in our results, which we had not considered. We adapted the suggested analyses (and those from the Bastiaansen et al., 2010 study) to our task and data structure (see new Figure 2 —figure supplement 1). In this analysis, we plotted β power as a function of position in the sequence for the five sequence types (all items InSeq, OutSeq item in position 2, OutSeq item in position 3,…). We did not find evidence of changes in β power with item position that would be indicative of a clear relationship with general expectancy (added to p6).

2. The authors need to integrate their findings with the large existing body of work that has been done investigating 20-40Hz β/low γ oscillations in the hippocampus and other brain regions during similar tasks. The finding currently feels isolated from previous work conducted in this realm, and integrating their findings with the existing literature would go a long way towards placing their findings in an existing framework and improving the impact of their work. The authors should read and cite the following studies at minimum (note that some are reviews that summarize many studies as a starting point, with a few particularly relevant original research papers listed below), integrating their results with previously published data and addressing leading theories regarding mechanisms and proposed functions. Knowledge of the below studies will provide a larger framework for their study, substantially increase the impact, and aid in the interpretation of results (particularly the few that also address 20-40Hz oscillations in the hippocampus).

Martin, Claire, Jennifer Beshel, and Leslie M Kay. 2007. "An Olfacto-Hippocampal Network Is Dynamically Involved in Odor-Discrimination Learning." Journal of Neurophysiology 98 (4): 2196-2205.

Berke, Joshua D, Vaughn Hetrick, Jason Breck, and Robert W Greene. 2008. "Transient 23-30 Hz Oscillations in Mouse Hippocampus during Exploration of Novel Environments." Hippocampus 18 (5): 519-29.

Iversen, John R, Bruno H Repp, and Aniruddh D Patel. 2009. "Top-Down Control of Rhythm Perception Modulates Early Auditory Responses."

Bastiaansen, Marcel, Lilla Magyari, and Peter Hagoort. 2010. "Syntactic Unification Operations Are Reflected in Oscillatory Dynamics during On-Line Sentence Comprehension." Journal of Cognitive Neuroscience 22 (7): 1333-47.

Weiss, Sabine, and Horst M Mueller. 2012. "'Too Many Betas Do Not Spoil the Broth': The Role of Β Brain Oscillations in Language Processing." Frontiers in Psychology 3 (January): 201.

Leventhal, Daniel K, Gregory J Gage, Robert Schmidt, Jeffrey R Pettibone, Alaina C Case, and Joshua D Berke. 2012. "Basal Ganglia Β Oscillations Accompany Cue Utilization." Neuron 73 (3): 523-36.

Kay, Leslie M. 2014. "Circuit Oscillations in Odor Perception and Memory." Progress in Brain Research 208 (January): 223-51.

Rangel, Lara M., Jon W Rueckemann, Pamela D. Riviere, Katherine R. Keefe, Blake S. Porter, Ian S. Heimbuch, Carl H. C.H. Budlong, and Howard Eichenbaum. 2016. "Rhythmic Coordination of Hippocampal Neurons during Associative Memory Processing." eLife 5 January 2016.

Trimper, John B., Claire R. Galloway, Andrew C. Jones, Kaavya Mandi, and Joseph R. Manns. 2017. "Γ Oscillations in Rat Hippocampal Subregions Dentate Gyrus, CA3, CA1, and Subiculum Underlie Associative Memory Encoding." Cell Reports 21 (9): 2419-32.

To clarify, our strategy in the first submission was to be conservative when comparing studies investigating “β oscillations” across behavioral states, tasks, species, and brain structures. This is because “β” frequencies vary across such studies, and presumably the etiology of the rhythm also varies across structures, so they may not all be examining the same β. Although this was a legitimate concern, in hindsight, we agree that we had been overly conservative and that our results were not properly contextualized. We have expanded the narrative on β throughout the paper (primarily in the results and discussion) and the findings are now better integrated with the existing literature (including the papers suggested above as well as others).

3. The authors averaged across four electrodes (spanning the proximal distal axis) for the analyses in Figure 4. It is unclear why the analysis in the latter figures was performed with an average for all electrodes across the proximodistal axis. Figures 1 and 2 show clear differences along the proximodistal axis. For example, in Figure 2 there appear to be significant changes in power as the animal learns in the most proximal electrode for theta and high γ, but this is not mentioned in the manuscript. It is unclear that this averaging is appropriate given the differences across the electrodes. The authors should perform these comparisons separately for proximal and distal electrodes at least. Importantly, if the authors analyze power changes in each location, are there changes in power that were missed by averaging across electrodes? Are there differences in oscillations between ABCDE and VWXYZ?

We apologize for not being clear on this. The trial-type analyses in Figure 4 were collapsed across electrodes for clarity, as the effects did not vary by electrodes (i.e., no significant interaction). This essentially amounts to showing the main effect of trial-type. We now show the effect of trial accuracy across electrodes in a new panel (Figure 4E) so the reader can get the sense that trial-type effects did not significantly vary across electrodes. We revised the other trial-type analyses (Figure 4A-C) to focus on data from the most proximal electrode, which showed the highest power, to avoid potential floor effects. However, the pattern is essentially the same for the other electrodes.

We now include a more thorough analysis of the session performance effects for theta and γ power (new Figure 3 —figure supplement 1). Unlike β, theta and (slow and fast) γ power does not increase with session performance (it either decreases or shows a nonlinear relationship). This is now included in the results and Discussion sections (p14-15).

We now show how β traces vary across electrodes and sequence (ABCDE in well-trained session; VWXYZ in Novel1 and 2 sessions) in the revised Figure 3B. While the general envelope is similar across sequences (sessions), β power in particular time epochs significantly varied with session performance.

4. The authors were very straightforward about their trial numbers in their supplementary data. It is unclear that there are enough trials for some of the comparisons. This is perhaps why some of the data was pooled in the manner shown in Figure 4. However, it is interesting to note that trends for individual rats do not necessarily follow the average. On that note, are parametric statistics appropriate for all comparisons?

We revised the trial-type analyses (Figure 4) to better account for the discrepancies in trial counts. First, we reformulated our ANOVA approach. We now use a linear mixed-effects model ANOVA (LMEM ANOVA) for all figures. This enables us to aggregate trial data from all animals while statistically accounting for inherent correlations in data from the same subject, as well as missing data and unequal trial counts. All data to which parametric statistics were applied met the normality assumption (Anderson-Darling test). Second, although we show the data and statistics for all trial-type comparisons for completeness, we now focus the trial-type analyses on one key contrast (InSeq+ vs OutSeq-). This contrast allows us to examine the effect of trial accuracy while controlling for the confounding influence of response duration (see point #5). This comparison is now performed using a more powerful analytical approach (random forest classifier; Figure 4D,E, Figure 4 —figure supplement 1; see point #9). The output data of the classifier analysis (Figure 4D) was not normally distributed, so non-parametric statistics were used. We have modified our methods and Results sections to reflect these changes.

Note that all analyses focus on group data aggregating all trials across all subjects (LMEM ANOVAs). The mean values from each rat are used for visualization only -- they are a bit variable, but that is not too surprising given that a few large values could drive the mean for one subject (but have little impact on the group mean). We did not observe significant anomalies in the distribution of values across trials.

5. It is unclear why the authors analyze data aligned to the end of the trial for several of their analyses (as opposed to the beginning of a trial). There is an assumption embedded in the decision that interesting oscillatory dynamics in both InSeq and Outseq trial types occur before the end of a poke period (as opposed to a decision point earlier). The authors could be missing important dynamics/differences at the beginning InSeq odor presentations, and should consider the alignment to the beginning of odor presentation as well. Perhaps a thorough analysis would report differences across running windows along the epoch (which would strengthen their argument that the differences are greater towards the middle or end of an interval).

Thank you for this helpful suggestion. Our goal in aligning the data to port withdrawal was to capture the peak in β power. But the reviewers are right – doing so obscured interesting dynamics about β power during odor presentations. Of particular interest is the fact that β gradually rises within odor trials and reaches a steady-state after ~500ms (see also points #1A and #6), which confounds the subset of analyses involving comparisons between trials of different durations (i.e., the original Figure 4). Analyses for that figure now show β traces aligned to both port entry and withdrawal to highlight these dynamics. Consequently, the corresponding results and discussion are now focused on trial-type analyses involving the most appropriate comparison, InSeq+ vs OutSeq- (which compares trial accuracy while matching for response duration), using data aligned to port entry (Figure 4D,E; Figure 4 —figure supplement 1).

Since the analyses of Figure 3 were not affected by this confound (because the analysis only included InSeq+ trials), we kept the original analyses (aligned to port withdrawal) but added mean β power traces to show the dynamics during odor trials (Figure 3B). However, we supplemented this initial analysis with the suggested analysis, by determining statistically significant time epochs during the odor periods using a cluster-based permutation approach (Figure 3B and Figure 5).

6. The authors indicate in multiple places that increases in the amplitude of 20-40Hz oscillations increase over the duration of the odor presentation/poke maintenance period. This is an interesting finding by itself and the authors should quantify this with statistics. A similar finding was reported previously in Rangel et al., 2016.

Again, thank you for bringing this up. As mentioned in point #5, we now show this increase by displaying mean β power traces during odor presentations (aligned to both port entry and withdrawal) and identified epochs in which power varied with session performance (Figure 3B, Figure 4A, and Figure 5).

7. A major problem with the presentation of the results is that there is no introduction to the task and the recordings. This is particularly important for the broad readership in eLife. As it is, the reader has to go to the methods where the explanation of the task is unclear. Furthermore, Figure 1 is not very informative and should be modified to convey information on the task, the fact that the time in the port is difference for InSeq and OutSeq, and the classification of trials as InSeq+, InSeq-, OutSeq+ and OutSeq-. Also, please provide an example of raw data recorded from one electrode and the Morlet wavelet analysis for that electrode. One would expect that the wavelet analysis would show clearly the theta phase aligned β (and γ?) bursts that become smoothed in the average data shown throughout the manuscript.

We appreciate the feedback on making our manuscript more accessible to a diverse readership. We added two paragraphs describing the task and recordings at the beginning of the Results section. We also modified Figure 1 to incorporate all requested changes: a more detailed schematic of the task, an experimental timeline, the trial-type classification, example trial traces and corresponding time-frequency analysis, and example behavioral data. The selected InSeq+ trial trace (Figure 1F, left) is suggestive of β bursts aligned to the theta phase, but this is not something we quantified in detail as we felt it was beyond the main scope of the manuscript.

8. Presentation of the behavioral outcome is lacking. How well did the animals perform in "lowest performance", "moderate performance" and "highest performance" in Figure 2? Please provide the data quantifying performance. Similarly, are the data in Table S4 statistically significant (as opposed to random)? Please show data for ABCDE and VWXYZ in the table.

A more detailed description of the behavioral measure, and of the performance in each of the three recorded sessions, has been added to the second paragraph of the results (instead of simply referring to our earlier work). It is important to note that our behavioral measure (sequence memory index; SMI) is essentially a modified Chi-Square test and thus considers the relative frequencies of each trial type within each animal separately (so it is not necessary to test whether distributions are different across subjects).

A more comprehensive account of trial counts is also now included in Table S1 (which replaces the old Table S4). Trial counts are shown for each subject, trial type, and session and is now separately showing total trials and trials included in the analyses (after artifact rejection). Note that some of the counts are different than in the original submission because artifact rejection is now based solely on activity in electrode 1 (the electrode used for the trial-type analyses in Figure 4).

9. The validity of the analysis of the data in Figure 4 showing that power varies across trial types is questionable. The authors had to merge and subsample the data. Did the authors attempt to decode trial types using machine learning or discriminant analysis? Perhaps this would provide more reliable information. Decoding analysis would allow calculation of decoding accuracy throughout the time course alleviating the problem with comparisons when the animal has left the port. Did the authors attempt to do this analysis using electrode locations that provided the largest differences in oscillatory power for the different trial outcomes?

Thank you for this alternate analysis suggestion. As mentioned in our response to point #4, the original subsampling approach in Figure 4 has been replaced by a linear-mixed effects model (LMEM) ANOVA. This approach, now used for all ANOVAs in the paper, allows the use of trials from all animals while statistically accounting for inherent correlations from the same subject as well as missing data and unequal trial counts.

Furthermore, as the reviewers suggested, the key trial-type comparison (InSeq+ vs OutSeq-) is now tested using a more powerful approach (random forest classifier; Figure 4D,E, Figure 4 —figure supplement 1). This analysis found a significant difference in β power between the two trial types.

10. In addition to p-values, please also consistently provide the statistic, degrees of freedom, and make sure that the number of animals tested is consistently clear.

We confirm that this information is now contained in the corresponding figure captions and Results sections. Note that the original supplementary tables (which only showed p values) have been removed after we revised our statistical approach.

One animal damaged its microdrive after the well-trained session, so the well-trained session includes data from five animals but the novel sessions from four animals. This information is now clarified in the second paragraph of the results, in the figure captions, and in Table S1.